# Learning What and Where: Disentangling Location and Identity Tracking Without Supervision

**Manuel Traub**
Neuro-Cognitive Modeling
University of Tübingen

**Sebastian Otte**
Neuro-Cognitive Modeling
University of Tübingen

**Tobias Menge**
Neuro-Cognitive Modeling
University of Tübingen

**Matthias Karlbauer**
Neuro-Cognitive Modeling
University of Tübingen

**Jannik Thümmel**
ML in Climate Science
University of Tübingen

**Martin V. Butz**
Neuro-Cognitive Modeling
University of Tübingen

## Abstract

Our brain can almost effortlessly decompose visual data streams into background and salient objects. Moreover, it can anticipate object motion and interactions, which are crucial abilities for conceptual planning and reasoning. Recent object reasoning datasets, such as CATER, have revealed fundamental shortcomings of current vision-based AI systems, particularly when targeting explicit object representations, object permanence, and object reasoning. Here we introduce a self-supervised LOCation and Identity tracking system (Loci), which excels on the CATER tracking challenge. Inspired by the dorsal and ventral pathways in the brain, Loci tackles the binding problem by processing separate, slot-wise encodings of 'what' and 'where'. Loci's predictive coding-like processing encourages active error minimization, such that individual slots tend to encode individual objects. Interactions between objects and object dynamics are processed in the disentangled latent space. Truncated backpropagation through time combined with forward eligibility accumulation significantly speeds up learning and improves memory efficiency. Besides exhibiting superior performance in current benchmarks, Loci effectively extracts objects from video streams and separates them into location and Gestalt components. We believe that this separation offers a representation that will facilitate effective planning and reasoning on conceptual levels. [1]

## 1 Introduction

Human perception is characterized by segmenting scenes into individual entities and their interactions [4; 13; 25]. This ability poses a non-trivial challenge for computational models of cognition [37; 84]: the binding problem [79]. Visual features need to be selectively bound into single objects, segregated from the background, and encoded by means of compressed stable neural attractors [5; 44; 46; 73].

Recent years have seen revolutionary progress in the ability of connectionist models to operate on complex natural images and videos [31; 50; 65]. Yet, neural network models still do not fully solve the binding problem [37]. Indeed, recent work on synthetic video-based reasoning datasets, like CLEVR, CLEVRER, or CATER [32; 43; 93], suggests that state-of-the-art systems [16; 89; 92] still struggle to model fundamental physical object properties, such as hollowness, blockage, or object permanence—concepts that children learn to master in the first few months of their lives [2; 63].

In a comprehensive review on the binding problem in the context of neural networks, Greff et al. [37] define three main challenges for solving the problem: *representation*, *segregation*, and *composition*. *Representation* refers to the challenge to effectively represent the essential properties of an object, including its appearance and potential dynamics. We will refer to these properties as the 'Gestalt' of an object [48; 86; 87]. Moreover, the individual objects' locations and motions dynamics should be

---

[1]Source Code: https://github.com/CognitiveModeling/Loci

disentangled from their Gestalt to enable compositional recombinations. Meanwhile, the representations should share a common format to enable general purpose reasoning. *Segregation* describes the challenge to extract particular objects from a perceived scene. This extraction should be done context- and task-dependently to identify the currently relevant entities. As a result, a good segregation should enable effective dynamic predictions of the whole, rather than only the parts. Finally, *composition* characterizes the challenge to develop object representations that enable meaningful re-combinations of object properties—particularly those that facilitate the prediction of object interaction dynamics. As a result, compositional representations enable conceptual reasoning about object properties as well as relations and interactions between objects.

We introduce a novel **Loc**ation and **I**dentity tracking system. While observing videos, Loci disentangles object identities ('what') from their spatial properties ('where') in a fully unsupervised, autoregressive manner. It is motivated by our brain's ventral and dorsal processing pathways [67; 80]. Loci's key contribution lies in how object-specific information is disentangled and recombined:

   (i) Loci fosters slot-respective object persistence over time via a novel combination of slot-specific input channels, temporal slot-interactive predictions via self-attention [83] followed by GateL0RD-RNN [39], and object permanence-oriented loss regularization.

  (ii) Our slot-decoding strategy combines object-specific Gestalt codes with parameterized Gaussians in a, to the best of our knowledge, novel manner. This combination fosters the emergent explication of an object's size, its position, and current occlusions.

 (iii) We improve sample and memory efficiency by training Loci's recurrent modules by means of time-local backpropagation combined with forward propagation of eligibility traces.

As a main result, we observe superior performance on the CATER benchmark: Loci outperforms previous methods by a large margin with an order of magnitude fewer parameters. Additional evaluations on moving MNIST, an aquarium video footage, and on the CLEVRER benchmark underline Loci's contribution towards the self-organized, disentangled identification and localization of objects as well as the effective processing of object interaction dynamics from video data.

## 2 RELATED WORK

Previous work by [59] has emphasized that, in general, unsupervised object representation learning is impossible because infinitely many variable models are consistent with the data. Inductive biases are thus necessary to ensure the effective learning of a system that segregates a visual stream of information into effective, compositional representations [37]. Accordingly, we review related work in the light of the binding problem and their relation to the proposed Loci system.

**Representation** A powerful choice of an encoding format is the formulation of 'slots', which share the encoding module but keep the codes of individual objects separate from one another. To ensure a common format between the slot-wise encodings, typically, slot-respective encoder modules share their weights [8; 60; 83]. To assign individual objects to individual slots, though, the system needs to break slot symmetry. Recurrent neural networks have been used to disentangle encodings or assignments [11; 28; 29; 35; 60]. Other mechanisms explicitly separate spatial slot locations [20; 42; 54], which we also do in Loci. However, instead of treating every spatial location as a potential object, each slot has a spotlight, which is designed to approximate the object's center. To further foster a compositional object representation, Loci enforces disentanglement of 'what' from 'where' by separating an object's Gestalt code—mainly representing shape and surface pattern—from its location, size (visual extent), and priority (current visibility with respect to other objects). This stronger disentanglement and more complex 'where' representation is related to work that models selective visual attention, realizing partially size-invariant tracking of one particular entity, such as a pedestrian [22; 45; 72]. Advancing this work, Loci tracks multiple objects in parallel, imposes interactive, object-specific spot-lights, and enables more compressed, object-specific appearance representations due to its novel way of combining 'what' and 'where' for decoding.

**Segregation** Segregating object instances from images is traditionally solved via bounding box detection [58; 74], where more advanced techniques extract additional masks for instance segmentation [15; 21; 40]. Through slot-attention mechanisms, recent unsupervised approaches partition

image regions into separate slots to represent distinct objects [36; 61]. To segregate objects from images, Burgess et al. [10] and von Kügelgen et al. [85] combine a soft attention (mask) approach with encoder, recurrent attention, and decoder modules, where slots compete for encoding individual objects. Loci pursues a similar approach but segregates objects even further to encourage object-respective 'what' and 'where' representations, where the latter additionally disentangle location, size, and approximate depth. Slot-respective masks compete via softmax attention to actively minimize the prediction error of the visual content. Moreover, a pre-trained background model separates potentially interesting from uninteresting regions. While we keep the background modeling rather simple in this work, more advanced techniques may certainly be applied in the near future [26; 68; 69; 82].

**Composition** Compositional reasoning in our model builds on two modules, which process object-to-object interactions and object dynamics. Object-to-object interactions are modelled using Multi-Head Self-Attention (MHSA) [83], in close relation to [27; 47]. Others have employed message passing neural networks to simulate object interactions [6; 18; 41]. Another promising approach uses Neural Production Systems to dynamically select a learnable rule to model object interactions [23]. Object dynamics are modelled using a recurrent GateL0RD module [39]. GateL0RD is designed to apply latent state updates sparsely, which encourages the maintenance of stable object encodings over long periods of time. Previous approaches have also employed recurrent structures to propagate slot-wise encodings forward in time [42; 49]. Although some previous works have combined recurrent structures with attention [33; 34], recent slot-attention frameworks tend to adopt fully auto-regressive designs based on transformers without explicit internal state maintenance [27; 47; 64].

**Tracking models** While our primary goal is to separate object location and Gestalt representations, extracted object locations likely facilitate object tracking; a wide area of research on its own [91]. State-of-the-art object tracking methods rely on features extracted via attention modules, which are typically applied autoregressively on individual frames [19; 64]. Again, these approaches do not explicitly foster a separation of 'what' from 'where', which may limit their applicability and accuracy in distractive environments, where humans still maintain high tracking skills, as shown in [55]. Loci maintains separate object encodings and thus copes with the presence of distractors more readily. The tracking model proposed in [55] is similar in spirit to Loci, extracting object encodings and propagating them with a competition mechanism and a recurrent module. However, to model complex objects their model relies on additional supervision. Disentangling movement and appearance is a common principle in video models with a notion of optical flow [57]. But optical flow based methods are not designed to deal with occlusions as they only represent parts of the scene that are currently visible. In contrast, Loci is able to maintain the encoding of an object even under occlusion.

**The CATER benchmark** We evaluate Loci mainly on the CATER challenge [32]. Previous SOTA methods on this challenge, like Multi-Hopper or OpNet [76; 96], have reused well-established neural network building blocks. Others have attempted to build their system on top of a supervised pre-trained bounding-box-based object detector [15; 97]. Most recently, current best results were reported using a video-level TFCNet [95], which makes efficient use of spatio-temporal convolutions on long videos. In contrast, Loci effectively combines an entity-oriented representational format with several kinds of interactive neural processing modules. While Loci disentangles location from identity when trained in a fully self-supervised manner, it can be further fine-tuned with supervision to minimize location-specific prediction error. Due to the employed inductive biases, Loci yields superior performance with a fraction of the number of parameters used in other models.

## 3 METHODS

The core idea of Loci involves two major aspects: First, distributing the visual input across recruitable slot-modules, which try to explain the ongoing scene dynamics in a compositional, mutually exclusive manner. A similar principle was shown to be a powerful inductive bias for emergent time series decomposition [70]. Second, splitting the encoding of a moving entity within a slot-module into its 'what' and 'where' components. These components are represented as a latent Gestalt vector ('what'), which can intuitively be understood as a symbolic encoding of an object's appearance, including its shape, color, size, texture, and other kinds of visual properties, and a latent spatial code ('where'), which explicates location, size, and relative priority [67; 80]. Moreover, we use the Siamese network approach implemented in slot-attention [60], sharing weights across slots. In fact, Loci employs the

same encoding and same decoding network for every slot starting from the raw image. Each encoder slot $k$ generates output Gestalt vectors $G_k$ and position codes $P_k$. Entity dynamics and interactions are then processed based on these pairs of latent Gestalt and location encodings. We predict the future states of latent encodings in a transition module and allow cross-slot interactions via multi-head self-attention. Figure 1 shows the main components of Loci. Information processing in one iteration unfolds from left to right. A detailed algorithmic description is provided in Appendix B. Here, we provide an overview and then detail the loss computation and training procedure.

## 3.1 LOCI OVERVIEW

The slot-wise encoder is based on a conventional ResNet architecture. The input consists of eight main features, which encode the image frame $I^t$ at current iteration $t$, the prediction error map $E^t$, a common background mask $\hat{M}_{bg}^t$ as well as, specifically for every slot $k$, a content mask $M_k^t$, a position map $\hat{Q}_k^t$, and maps for both the other masks $\hat{M}_k^{t,s}$ and the other positions $\hat{Q}_k^{t,s}$ (cf. Figure 1). The computation path of Gestalt and position for an individual slot is tree-shaped, starting with a shared ResNet-trunk after which the path is split into four pathways, which compute Gestalt, location, size, and priority. This encoder design encourages that separately moving entities are encoded in individual ResNet activity dynamics as well as that each slot encodes 'what' (Gestalt) and 'where' (location, size, and priority) of the slot-encoded moving entity separately but conjointly. Thereby, the error map $E^t$ encourages active error minimization, following the principle of predictive coding fostering attractor dynamics [44; 73]. Appendix B specifies further details.

The transition module then predicts location $P_k^t$ and Gestalt dynamics $G_k^t$ as well as interactions between slots. The module is loosely based on the architecture of a transformer-like encoder [83], where several multi-head attention layers are stacked together with residual feed forward layers in

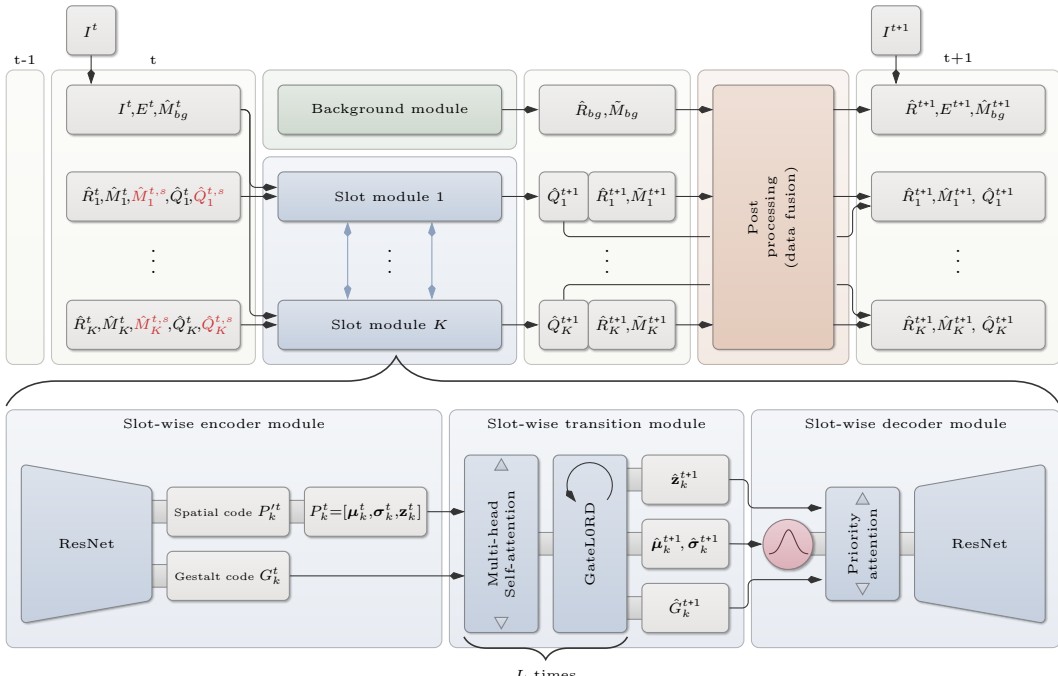

Figure 1: Loci's main processing loop has three components: Individual entities $k$ are processed by a *slot-wise encoder module* to generate the disentangled latent Gestalt code $G_k$ and position code $P_k$. The *transition module* consists of $L$ alternating blocks of multi-head self-attention (modeling slot-to-slot interactions) and GateL0RD (predicting within-slot dynamics). The resulting Gestalt and position codes are finally combined by the *slot-wise decoder* into entity-respective RGB reconstructions $R_k$ and masks $M_k$. Tensors colored in red ($\hat{M}_k^{t,s}$, $\hat{Q}_k^{t,s}$) are calculated as complements from $\hat{M}_k^t$ and $\hat{Q}_k^t$.

between. We replace the residual feed forward layers with residual GateL0RD [39] layers. GateL0RD is a recent gated recurrent neural network module, which is very well-suited to learn and distinctively encode interaction events, which have been described as "structured, describable, memorable units of experience" [3], cf. [94]. In accordance with our entity-focused processing approach, we apply a Siamese GateL0RD version, operating on the individual slot level while receiving information form other slots via the attention layers. At the end of the transition module, the Gestalt code is pushed through a binarization layer, inspired by the principle of vector quantization [81]. This layer enforces an information bottleneck and thus contributes to the development of disentangled entity codes.

The slot-wise decoder recombines the 'what' and 'where' output from the transition module. To do so, first the potential influence of each Gestalt code $G_k$ is computed over the output range by means of a 2d isotropic Gaussian parameterized by the position code $P_k$, yielding density maps $Q_k$. Next, a priority-$\hat{z}$-based attention is applied to account for the fact that only one slot-object can be visible at any location (transparent objects are left for future work, see Algorithm-2 in Appendix B). As a result, when two slots cover the same location, the one with the lower priority will have its feature maps set to zero. The rest of the decoder is based on a conventional ResNet, which increases the resolution back to the video size. Like the encoder, the decoder shares weights across all slots. Eventually, the decoder outputs predictions of RGB entity reconstructions $\hat{R}_k^{t+1}$ and individual mask predictions $\tilde{M}_k^{t+1}$ for each slot $k$. The masks from all slots and from the background are then combined to construct the final output image prediction $\hat{R}^{t+1}$ and competitive mask predictions $\hat{M}_k^{t+1}$. To fully reconstruct the image, Loci uses its simple background module, which generates background image estimates $\hat{R}_{bg}$ and a background mask $\tilde{M}_{bg}$. In case of a static background, a Gaussian mixture model over the whole training set is used and a flat background mask set to the bias value $\theta_{bg}$. For more dynamic backgrounds, we employ a simple auto encoder.

## 3.2 Training

Loci is trained using a binary cross-entropy loss ($L_{BCE}$) pixel-wise on the frame prediction applying rectified Adam optimization [RAdam, cf. 56]. Several regularizing losses are added to foster object permanence. Additionally, to speed-up learning, we use truncated backpropagation trough time and enhance the gradients with an e-prop-like accumulation of previous neural activities [7].

Empirical evaluations showed that backpropagating the gradients through the networks' inputs creates instabilities in the training process. Thus, we detach the gradients for the latent states. As a result, the only part of the network that needs to backpropagate gradients trough time are the GateL0RD layers. Here, we found that using the described combination of e-prop and backpropagation is not only comparable in terms of network accuracy, it also greatly decreases training wall-clock time, as it allows the use of truncated backpropagation with length 1, effectively updating the weights after each input frame (see supplementary material for details).

Another important aspect for successful training is the use of a warmup phase, where we mask the target of the network with a foreground mask computed with a threshold $\tau$:

$$M = \tau < \texttt{Mean}\left(\left(I^t - \hat{R}_{bg}\right)^2, axis = \text{'rgb'}\right), \tag{1}$$

where $\hat{R}_{bg}$ is the background model estimate detailed above. A black background is used instead of $\hat{R}_{bg}$ to construct the next frame prediction during this phase. The foreground mask together with the zeroed background enforces the network to focus on reconstructing the foreground objects only. This encourages the usage of the position-constrained encoding and decoding mechanism. After about 30 000 updates—when the network has sufficiently learned to use the position encodings—we switch from the masked foreground reconstruction to the full reconstruction.

**Object Permanence Loss**  To encourage object permanence, an additional loss is computed based on Equation 2, which favors a slot that keeps encoding the same object, even if the object is temporarily occluded and thus invisible:

$$L_o = \sum_k \left(\mathfrak{D}_k(P_k^t, \hat{G}_k^t) - \mathfrak{D}_k(P_k^t, G_k^t)\right)^2, \tag{2}$$

$$\hat{G}_k^t = \hat{G}_k^{t-1}(1 - max(M_k^t)) + G_k^t max(M_k^t), \tag{3}$$

where $P_k$ and $G_k$ denote location and Gestalt encoding in slot $k$, $\mathfrak{D}_k$ refers to the RGB part of the decoder network, while $\hat{G}_k^t$ denotes the Gestalt code averaged around the last time step in which the entity was visible, and $M_k^t$ denotes the mask of slot $k$ at time step $t$. As a result, $L_o$ is only applied when the object becomes invisible, which is the case when $M_k^t$ is approaching zero.

**Time Persistence Loss**  A second mechanism to enforce object permanence and to also regularize the network towards temporal consistent slot encodings, is a time persistent regularization loss:

$$L_t = 0.1 \sum_k \left( \mathfrak{D}_k(p_0, G_k^{t-1}) - \mathfrak{D}_k(p_0, G_k^t) \right)^2 , \tag{4}$$

where again $\mathfrak{D}_k$ refers to the RGB part of the decoder network and $p_0$ is the center position in the image. $L_t$ essentially penalizes large visual changes in the decoded object between two consecutive time steps.

**Position Change Loss**  In order to encourage the network to predict small slot-position changes over time, a simple $L_2$ regularization loss is based on the position change between two time steps:

$$L_p = 0.01 \sum_n (P_k^{t-1} - P_k^t)^2 \tag{5}$$

**Supervision Loss**  Finally, for the experiments that are using a supervised target object location signal for fine-tuning, detailed in Equation 6, we added a gating network $\Phi$ that operates on the latent Gestalt codes $G_k^t$ before binarization and predicts a single softmax probability, which is used to decide which entity corresponds to the Snitch in the CATER dataset. The location of the selected entity is then used in an $L_2$-loss to foster regression to the target location provided in the dataset.

$$L_s = \mu_s \left( \sum_k P_k^t \Phi(\hat{G}_k^t) - p_{snitch}^t \right)^2 \tag{6}$$

The final loss for the network results from adding the individual loss components, denoting the unsupervised and supervised losses, respectively:

$$L_{unsup} = L_{BCE} + L_o + L_t + L_p, \tag{7}$$
$$L_{sup} = L_{BCE} + L_o + L_t + L_p + L_s. \tag{8}$$

## 4 EXPERIMENTS & RESULTS

We performed unsupervised training in all experiments. Only for the CATER challenge we additionally evaluated a version where we fine-tuned the network via supervision using Equation 6.

### 4.1 CATER-TRACKING CHALLENGE

In the CATER tracking challenge (cf. Figure 2), the task is to locate a unique yellow object, called Snitch, at the end of a video sequence. During the video, different geometric objects rotate, move,

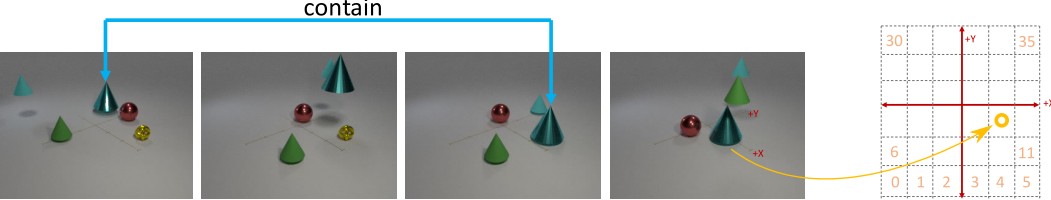

Figure 2: The CATER Snitch localization challenge: The task is to locate the yellow spherical object called Snitch within the last frame. The challenge is that the Snitch might be contained and moved by a cone. So its location has to be inferred by recognizing and remembering a containment event and then tracking the position of the container. Image adapted from [32].

Table 1: Quantitative results of the CATER Snitch localization challenge. Referenced results are from [96] and from the other system-respective papers.

| Method | Parameters (M) | Top 1 | Top 5 | $L_1$ (grid) | $L_2$ |
|---|---|---|---|---|---|
| Random | - | 2.8 | 13.8 | 3.9 | |
| Transformer [83] | 15.01 | 13.7 | 39.9 | 3.53 | |
| SINet [62] | 138.69 | 21.1 | 47.1 | 3.14 | |
| TSN (RGB) + LSTM [88] | | 25.6 | 67.2 | 2.6 | |
| DaSiamRPN [97] | | 33.9 | 40.8 | 2.4 | |
| I3D-50 + LSTM [16] | | 60.2 | 81.8 | 1.2 | |
| Hopper-transformer [96] | 15.01 | 61.1 | 86.6 | 1.42 | |
| TSM-50 [53] | | 64.0 | 85.7 | 0.93 | |
| TPN-101 [92] | | 65.3 | 83.0 | 1.09 | |
| Hopper-sinet [96] | 139.22 | 69.1 | 91.8 | 1.02 | |
| Inferno [17] | - | 71.7 | 88.9 | - | |
| Hopper-multihop [96] | 6.39 | 73.2 | 93.8 | 0.85 | |
| Aloe [24] | | 74.0 | 94.0 | 0.44 | |
| OPNet [76] | - | 74.8 | - | 0.54 | |
| TFC V3D Depthwise [95] | 24.64 | 79.7 | 95.5 | 0.47 | |
| **Loci supervised (ours)** | **2.96** | **90.7** | **98.5** | **0.14** | 0.14 |
| Loci unsupervised (ours) | 4.14 | 78.4 | 92.0 | 0.45 | 0.44 |

frame 191      frame 194      frame 287      frame 291

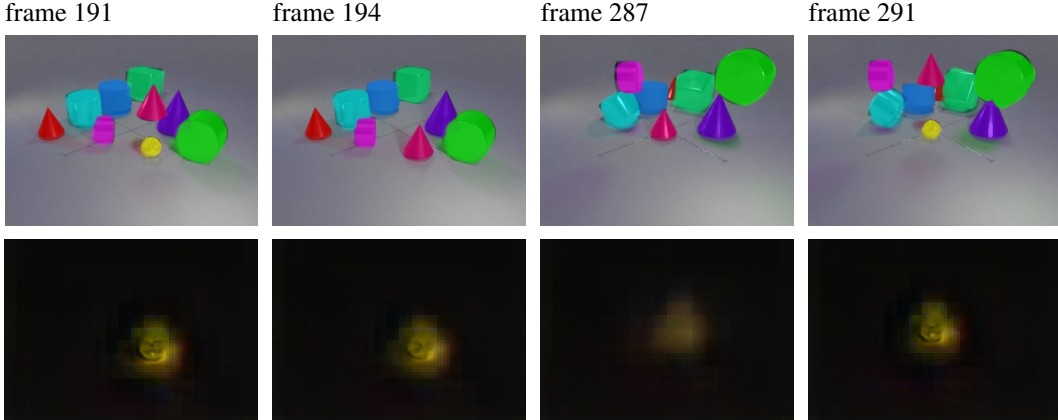

Figure 3: Object permanence shown on a CATER tracking example. While the network (after completely unsupervised training) struggles to keep the shape of the Snitch when contained for a longer time span, color and, importantly, the position of the Snitch are preserved during containment. **Top row**: tracked objects visualized trough colored masks. **Bottom row**: RGB representation of the Snitch.

and hop in a scripted random way. By doing so, cone objects can engulf other objects and move them to another location before releasing them again, which can lead to situations where the Snitch remains hidden in the last frame of the video. Therefore, the challenge is not only to *recognize containment* events, but also to *track* the specific cone that contains the Snitch. For classification purposes, the 3D Snitch position is partitioned into a $6 \times 6$ grid, resulting in a total of 36 classes. In order to account for the fact that a small location error could lead to miss-classification, a common reported metric is the $L_1$ grid distance. We additionally report the continuous $L_2$ distance with grid length 1, such that the distance is comparable to, but more informative than the $L_1$ grid distance.

As described in Girdhar & Ramanan [32], we split the dataset with a ratio of 70:30 into a training and test set and further put aside 20% of the training set as a validation set, leaving 56% of the original data for training. To blend-in the supervision loss, we first set the supervision factor $\mu_s = 0.01$ for the first 4 epochs, fostering mostly unsupervised training, and then set $\mu_s = 0.1$ for the duration of the training process and to $\mu_s = 0.\overline{3}$ during the last epochs, to give the Snitch location a weak pull towards the target location.

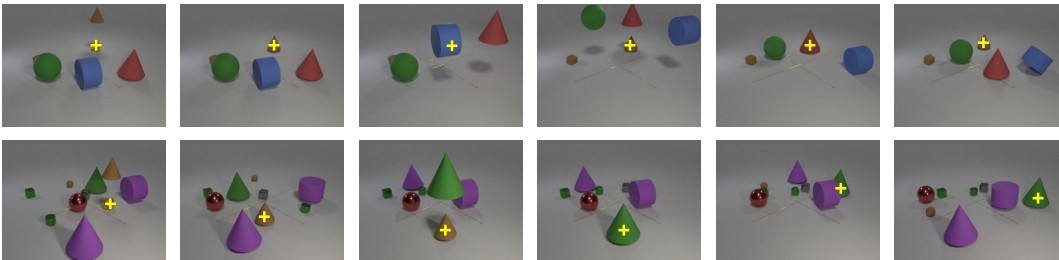

Figure 4: Two challenging CATER tracking examples with several co-occurring containments / occlusions: Video frames overlaid with the predicted Snitch location (network trained with supervision).

In order to produce labels from the purely unsupervisedly trained network, we train a separate small classifier with around 17k parameters, which only operates on the latent states of the trained Loci network with a correction and a gating network. The correction network computes a residual for the location of each entity. The gating network computes softmax probabilities, which are used to select the location belonging to the Snitch object similar to Equation 6. To prevent supervised gradient flow, the classifier network module is trained once the unsupervised training has finished. Eventually, the location and Gestalt codes are computed for the whole dataset and extracted into a separate data file, from which the classifier module is trained, again using a 70:30 train/test split.

As shown in Table 1, Loci not only surpasses all previous methods by a large margin—achieving a top 1 accuracy of 90.7% and a top 5 accuracy of 98.5% with an $L_1$ distance of 0.14—it also surpasses most of the existing work when the main architecture is trained purely unsupervised. Loci learns object permanence merely by means of inductive bias and regularization. This is shown in Figure 3, where Loci keeps the representation of the contained Snitch in memory. While the shape details blur over time, importantly, the locating is correctly tracked to reconstruct the Snitch once it is revealed. Figure 4 also demonstrates the tracking performance in case of several co-occurring occlusions and double contained situations, where two cones are covering the Snitch. Here, the location output of the supervised gating network is marked in the video frames.

In additional studies (detailed in Appendix A.3.5), we furthermore investigate Loci's object permanence and also demonstrate the effective disentanglement of position, Gestalt, size, and priority codes, which underlines Loci's ability to separate position and Gestalt of objects from a scene without supervision.

## 4.2 MOVING MNIST

The moving MNIST (MMNIST) challenge is a dataset for video prediction [77]. The task is to predict the motion of two moving digits, with a maximum size of $28 \times 28$ pixels, moving independently inside a $64 \times 64$ pixel window and bouncing off the walls in a predictable manner. While the dataset is usually implemented by simply bouncing the entire $28 \times 28$ MNIST sub-image within the $64 \times 64$ window, we first crop each digit to its actual appearance in order to obtain a realistic bounding box of the digit and thus to generate more realistic bouncing effects (once the actual number touches the border, instead of its superficial bounding box). This removes the undesired bias in the dataset to remember each digit's individual bounding box, individually, which is then the only way to correctly predict a bounce off a wall. Instead, now the network can predict the bounce based on the pixel information alone.

While the task was originally formulated to predict the next 10 frames after receiving the first 10 frames, we also evaluate the ability to generate temporal closed-loop predictions for up to 100 frames after being only trained to predict 10 frames. We compare Loci to the state-of-the-art approach PhyDNet, also designed to disentangle 'what' from 'where'; more specifically: Physical dynamics from unknown factors such as the digit's Gestalt code. While using the original code provided by the authors, we did not reach the same reported performance of PhyDNet using our unbiased MMNIST dataloader. Nevertheless, PhyDNet reaches a high accuracy for the 10 frame prediction, which is only slightly topped by Loci. For extended temporal predictions, though, PhyDNet quickly dissolves the digits, while Loci preserves the Gestalt of each digit over the 100 predicted frames, as

Table 2: Moving MNIST prediction accuracy PhyDNet vs Loci. Both networks where trained using the same dataloader to predict the next 90 frames given an input sequence of 10 frames.

| Method | $T = 10$ | | $T = 20$ | | $T = 30$ | | $T = 50$ | | $T = 100$ | |
|---|---|---|---|---|---|---|---|---|---|---|
| | MSE | SSIM | MSE | SSIM | MSE | SSIM | MSE | SSIM | MSE | SSIM |
| PhyDnet [38] | 35.6 | 0.913 | 58.6 | 0.830 | **80.7** | 0.739 | **113.9** | 0.594 | **155.1** | 0.484 |
| Loci | **30.7** | **0.923** | **52.5** | **0.881** | 98.3 | **0.814** | 165.1 | **0.720** | 221.2 | **0.639** |

Figure 5: Comparison between ground truth **top row**, PhyDNet **center row** and Loci **bottom row** for a prediction of up to 90 frames. Both PhyDNet and Loci were trained on 10 frame prediction. While in PhyDNet, the appearance of the digits dissolves after a few frames beyond the initial training distribution, Loci manages to keep the Gestalt code and the location accurate until the fourth collision at around frame 42. The Gestalt codes remain stable until the end of the considered 100 time steps.

shown in Figure 5. Table 2 shows a qualitative comparison between Loci and PhyDNet: While Loci consistently outperforms PhyDNet for the structural similarity index measure (SSIM, [90]), PhyDNet has a lower MSE than Loci after 30 frames. This might be due to the blurring of PhyDNet, which accounts for the uncertainty in the position estimate at the cost of the digits shape.

## 4.3 OTHER DATASETS & SUPPLEMENTARY MATERIAL

Apart from a video footage for the main experiments and further algorithmic details on Loci, we provide additional insights and tests in the supplementary material: We evaluate the real world tracking performance of Loci on a ten hour aquarium footage found on YouTube Additionally, we examine Gestalt preservation and indicators of intuitive physics in closed loop predictions on the CLEVRER dataset [93]. Furthermore, several ablation studies are provided.

## 5 CONCLUSION AND FUTURE WORK

We presented Loci—a novel location and identity disentangling artificial neural network, which learns to process object-distinct position and Gestalt encodings. Loci significantly exceeds current state-of-the-art architectures on the CATER challenge, while requiring fewer network parameters and less supervision. Loci thus offers a critical step towards solving the binding problem [37; 79; 84]. We particularly hope that the mechanisms proposed in this work bear potential to enrich the field of object representation learning and highlight the importance of well-chosen inductive biases.

Currently, Loci operates on static background estimates and cameras, which we denote as its main limitation. We intend to extend the background model to incorporate rich and varying backgrounds more flexibly in future work. Potential avenues for this extension may include an explicit, separate encoding of ego-motion and depth [52]. Furthermore, image reconstructions may be fused with subsequent image inputs in an information-driven manner. We also expect to create even more compact object-interaction encodings following theories of event-predictive cognition and related conceptions and models in computational neuroscience [12; 14; 30; 78]. Moreover, we are excited to explore the potential of Loci to address problems of intuitive physics and causality [66; 71; 75], seeing that Loci offers a suitable compositional structure [37; 51] to enable symbolic reasoning. Finally, we hope that Loci will also be well-combinable with reinforcement learning and model-predictive planning in system control setups to pursue active, goal-directed environmental interactions.

## 6 ACKNOWLEDGEMENT

This work received funding from the Deutsche Forschungsgemeinschaft (DFG, German Research Foundation) under Germany's Excellence Strategy – EXC number 2064/1 – Project number 390727645 as well as from the Cyber Valley in Tübingen, CyVy-RF-2020-15. The authors thank the International Max Planck Research School for Intelligent Systems (IMPRS-IS) for supporting Manuel Traub and Matthias Karlbauer, and the Alexander von Humboldt Foundation for supporting Martin Butz and Sebastian Otte. We also thank Simon Frank for the Gestalt-Code analysis in Figure 16.

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

## A APPENDIX

As additional content, we first provide details on further evaluations we conducted with Loci. In order to evaluate the tracking performance of Loci in a real world example, we trained it on a 10 hour aquarium footage found on YouTube. The task poses the additional challenge to cope with backgrounds that are not fully stationary. The results are shown in Figure 6 and demonstrate that Loci is able to track 15+ objects in a complex real world environment.

The Gestalt preserving performance of Loci for closed loop predictions are also demonstrated exemplary on the CLEVRER dataset in Figure 7. Here, the effects of collisions of different geometric objects are predicted into the future. While the location deviates visibly over time, Loci is able to preserve the Gestalt code also for more complex objects in a closed loop setup, which is considered specifically challenging for RNNs.

### A.1 CATER EVALUATION DETAILS

In order to produce the results from Table 1 we trained five networks with different initial seeds and and then evaluated each network five times with different initial seeds. In Table 1 we reported the mean results over the five evaluation runs from our best performing network (Network 3 from Table 3).

### A.2 BACKGOUND MODEL

In order to compute an background model for datasets with a simple static background like CATER, CLEVR or CLEVRER we use a simple Gaussian Mixture Model. The specific function used is *createBackgroundSubtractorMOG2* from OpenCV [9] which we use with a learningrate of 0.00001 to compute an background image based on the training set.

For more complex backgrounds like in the aquarium example where the camera is still static, we use an autoencoder ResNet that encodes the input image into a latent vector with the same size as the Gestalt code. This latent background code is then run through an residual GateL0rd in order to capture temporal dynamics, like the chancing water level, and is then run through an ResNet decoder. The background autoencoder is pretrained using a L1 reconstruction loss in order to focus on the most dominant features of the background and to not care so much about foreground objects.

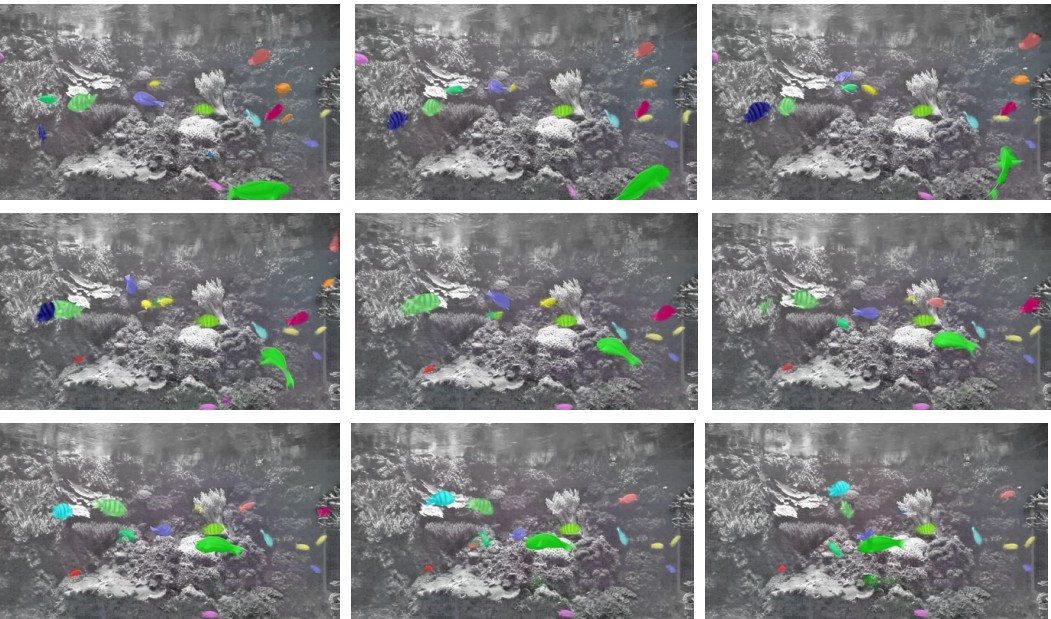

Figure 6: Fully unsupervised real world tracking example trained on 10 hour aquarium footage.

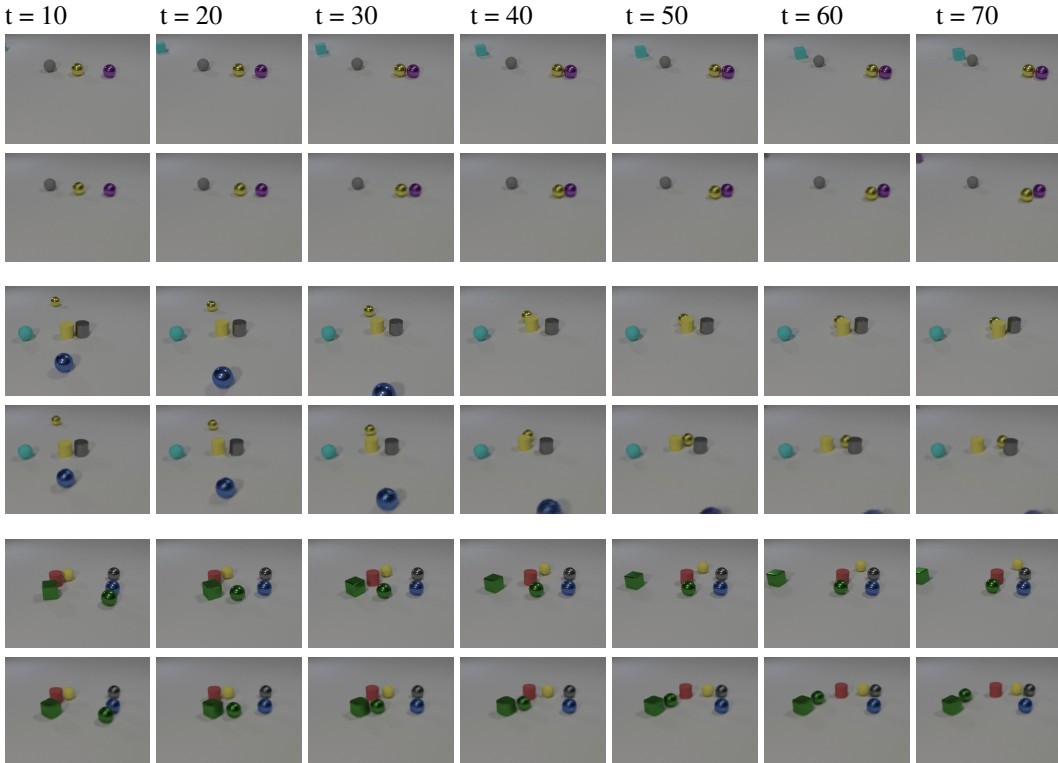

Figure 7: Three examples of predicting object dynamics from the CLEVRER[93] benchmarks. Loci runs 70 time-steps in closed loop after 50 times-steps of teacher forcing. Top rows show ground truth wile bottom rows show Loci's closed loop predictions.

## A.3   ABLATIONS

Using the binary cross-entropy with non binary targets, while producing valid gradients, gives little insights into the network's actual performance. In order to better compare different designs and to also take the objects into focus, we use a modified $L_2$-loss for our ablation studies that is computed based on Equation 9:

$$L2_{object} = \sum \left( (O^t_{b,c,h,w} - I^{t+1}_{b,c,h,w}) \sqrt{\sum_{c=1 \to 3} \frac{1}{3} \left( I^{t+1}_c - B^{t+1}_c \right)^2} \right)^2 \tag{9}$$

Here the MSE is masked with the error between the input and the background. As a result, we get a much higher error when the network produces a tracking error compared to a background reconstruction error. The $L2_{object}$ loss takes into account instances, where, for example, the network overlooks an object or, makes a mistake in the prediction of the movement of an object. It thus offers itself as a good metric for comparing the performance of different architectures during training.

### A.3.1   GATEL0RD VS LSTM

As shown in Figure 8, using GateL0RD within the predictor network significantly increases the $L2_{object}$-Loss. Thus, it appears that GateL0RD's piece-wise constant latent state regularization mechanism indeed suitably biases the network towards assuming object permanence.

### A.3.2   INPUT CHANNELS

In Figure 10, we show the networks performance in terms of $L2_{object}$ when we zero out different input channels.

Table 3: Evaluations on the CATER Snitch challenge for supervised training. Table shows five networks trained with different initial seeds, each evaluated five times with different initial seeds.

| Network | Top 1 | Top 5 | $L_1$ (grid) | $L_2$ |
|---------|-------|-------|--------------|-------|
| 1 | 91.7 | 99.0 | 0.118 | 0.122 |
|   | 90.8 | 98.8 | 0.129 | 0.126 |
|   | 91.1 | 98.8 | 0.119 | 0.121 |
|   | 91.4 | 98.9 | 0.122 | 0.123 |
|   | 91.3 | 98.8 | 0.123 | 0.126 |
| 2 | 91.1 | 98.6 | 0.129 | 0.133 |
|   | 90.4 | 98.3 | 0.144 | 0.141 |
|   | 89.8 | 98.3 | 0.151 | 0.144 |
|   | 90.0 | 98.5 | 0.149 | 0.140 |
|   | 89.5 | 98.2 | 0.164 | 0.151 |
| 3 | 89.6 | 97.8 | 0.173 | 0.170 |
|   | 90.1 | 98.0 | 0.159 | 0.157 |
|   | 89.5 | 97.9 | 0.174 | 0.164 |
|   | 88.9 | 97.5 | 0.174 | 0.167 |
|   | 89.8 | 98.1 | 0.162 | 0.157 |
| 4 | 91.5 | 99.1 | 0.112 | 0.118 |
|   | 90.4 | 98.9 | 0.133 | 0.128 |
|   | 91.0 | 98.7 | 0.125 | 0.129 |
|   | 90.5 | 99.1 | 0.125 | 0.124 |
|   | 90.8 | 98.8 | 0.127 | 0.126 |
| 5 | 91.7 | 98.2 | 0.130 | 0.139 |
|   | 91.4 | 98.4 | 0.135 | 0.139 |
|   | 91.2 | 98.0 | 0.146 | 0.150 |
|   | 91.8 | 98.3 | 0.130 | 0.140 |
|   | 91.3 | 98.1 | 0.150 | 0.151 |

Table 4: Evaluations on the CATER Snitch challenge for supervised training, statistics of the 25 evaluations presented in Table 3

| Metric | min | mean | std | max |
|--------|-----|------|-----|-----|
| Top1 | 88.9 | 90.7 | 0.8 | 91.8 |
| Top5 | 97.5 | 98.5 | 0.4 | 99.1 |
| $L_1$ (grid) | 0.112 | 0.140 | 0.019 | 0.174 |
| $L_2$ | 0.118 | 0.139 | 0.015 | 0.170 |

### A.3.3 REGULARIZATION LOSSES

In Figure 11, we show the networks performance in terms of $L2_{object}$ without certain regularization losses.

### A.3.4 E-PROP VS BACK-PROPAGATION THROUGH TIME

In Figure 9 we compared the $L2_{object}$ loss for truncated back-propagation through time with different sequence lengths against a version where we used back-propagation together with e-prop. In all experiments the networks are trained on the full Cater sequence with 300 frames, which are fed into the network sequentially. We then updated and afterwards detached the gradients using truncated back-propagation through time with different time intervals. Using e-prop informed gradients not only drastically sped up training time, but it was also more sample-efficient while achieving the same performance as truncated back-propagation through time with an interval of three time steps. Especially interesting is the experiment where we tested truncated back-propagation with a sequence length of one, without using e-prop's informed gradients. In this case, the performance is significantly worse than when using e-prop's informed gradients.

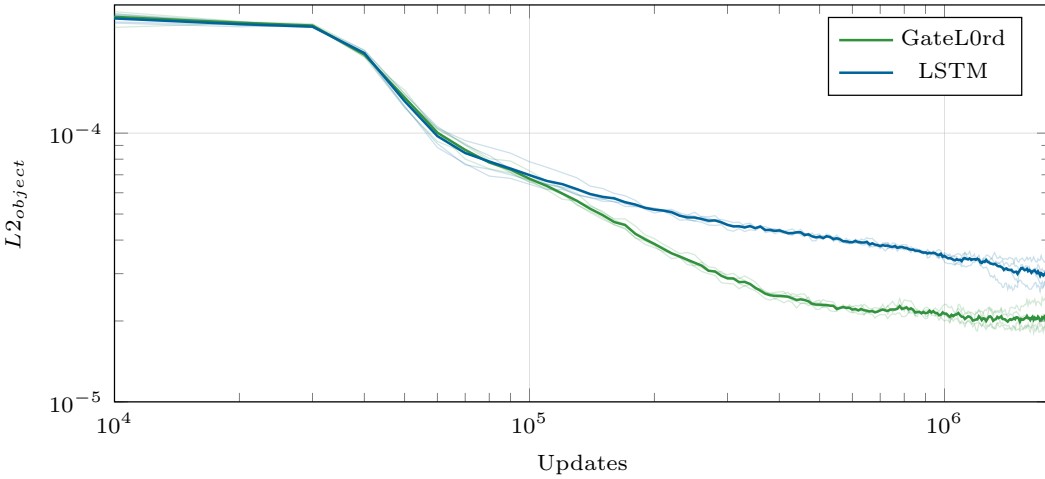

Figure 8: Comparison between GateL0RD and LSTM modules within the predictor part of the network.

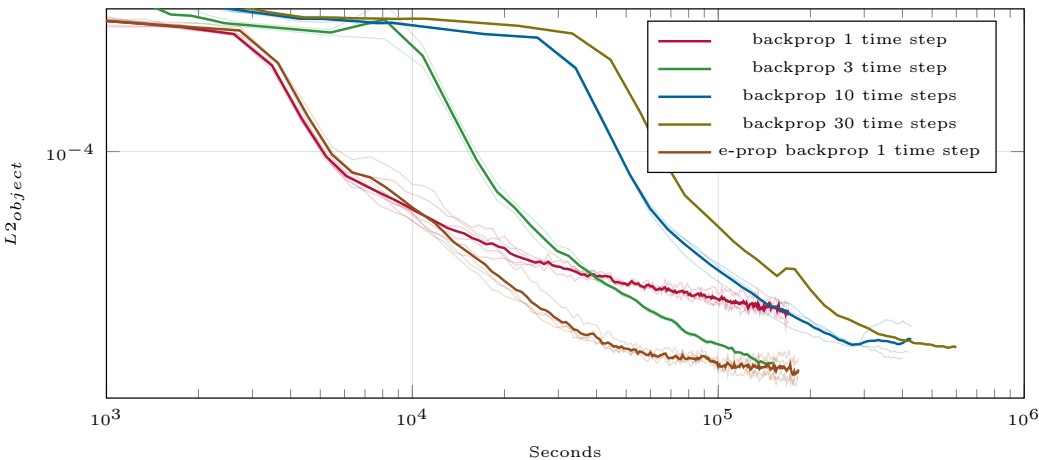

Figure 9: Comparison between different sequence lengths for truncated back-propagation trough time vs back-propagation and e-prop.

### A.3.5 LESION STUDIES

In order to quantify the effect of the Gestalt and position codes, we conducted an ablation by adding normally distributed noise with standard deviations $\sigma \in [0, 0.1, ..., 1.0]$ to the Gestalt and position codes provided by the encoder–right before they were forwarded to the transition module—and calculated the resulting accuracies and errors, which are reported in Figure 12 and Figure 13 for the CATER and moving MNIST benchmarks, respectively. Furthermore, exemplary images of according position, priority, size, and Gestalt code manipulations are presented in Figure 14 and Figure 5, again for the CATER and moving MNIST benchmarks. Effectively, manipulating the according codes result in changes in position, priority, size, or Gestalt code, demonstrating the disentangled encoding of object features in the respective latent codes. In particular, while the Gestalt-code manipulation results in changes of the actual numbers in the moving MNIST experiment under conservation of positions (first vs. third row of Figure 5), the position-code manipulation alters the position of the numbers in space without modifying the object shape (first vs. fourth row of Figure 5).

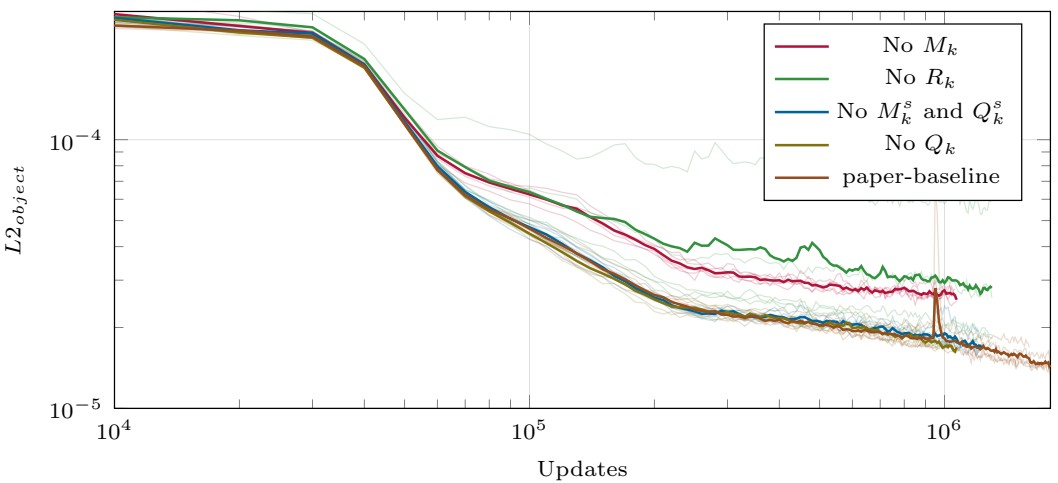

Figure 10: Importance of different input channels.

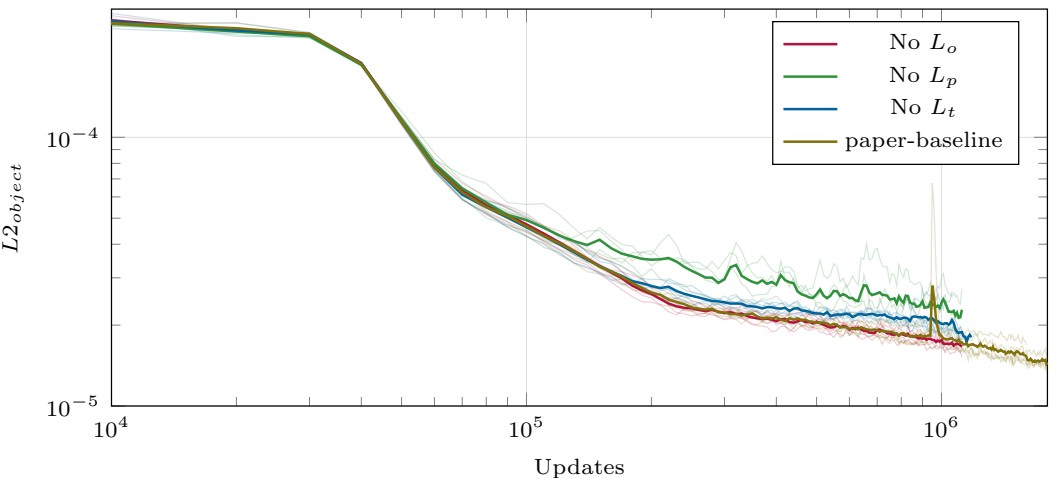

Figure 11: Importance of regularization losses.

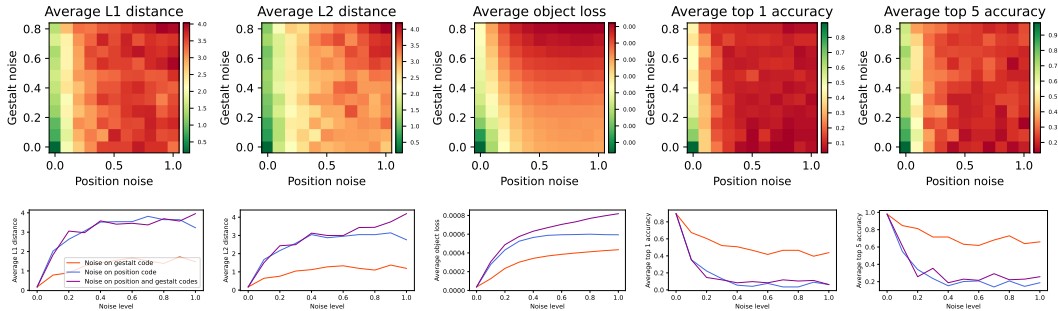

Figure 12: Gestalt-position lesion study on the CATER benchmark. Top: Metrics resulting from different *noise* intensities *added* to the *Gestalt* and *position codes*. Bottom: Explicit visualization of the bottom-most row (manipulating position only, blue), left-most column (manipulating Gestalt only, orange), and diagonal from bottom left to top right in the plots at the top (manipulating both position and Gestalt, purple).

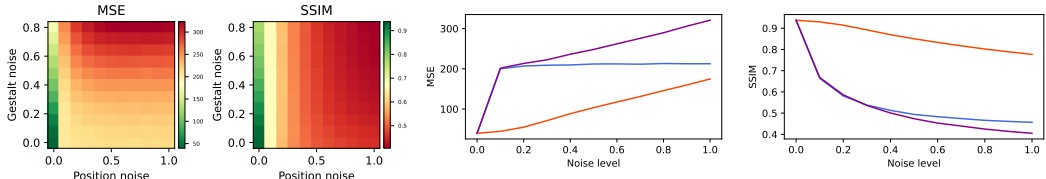

Figure 13: Same analysis as Figure 12, applied to the moving MMNIST dataset.

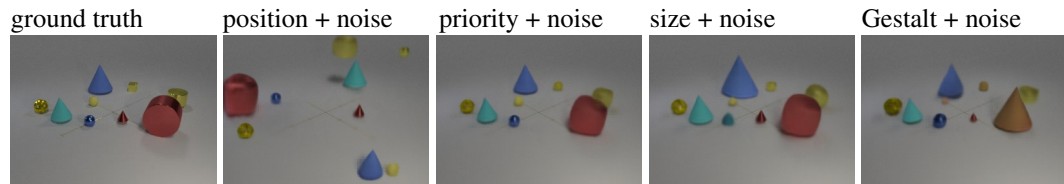

Figure 14: To study the disentanglement of the different parts of our learned codes, we perform a lesion study. The principle of the study is as follows: For each component code (position, priority, size, and Gestalt) we add gaussian noise to the code before applying the decoder while keeping all other codes as predicted by the model.

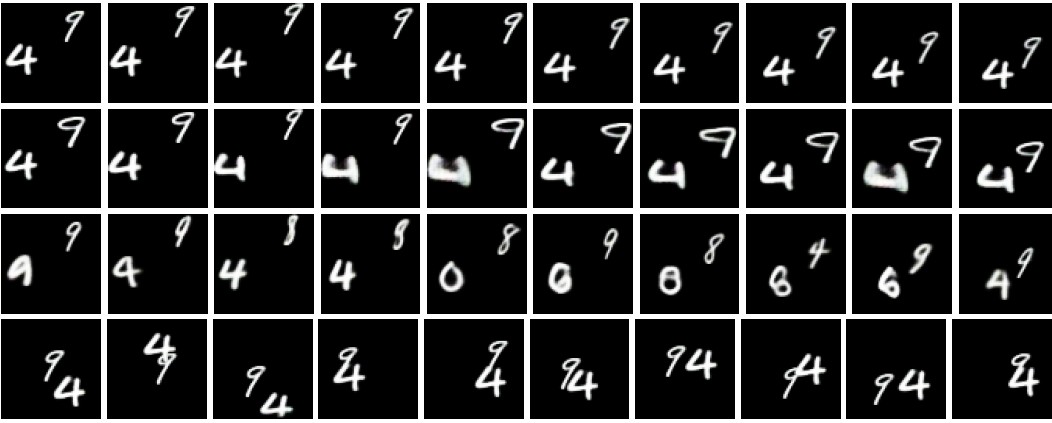

Figure 15: Lesion study on the moving mnist closed loop predictions (frames $t = 11$ to $t = 20$). **First row:** undisturbed prediction. **Second row:** $0.25$ scaled gaussian noise added to the position (size) code, which is then cliped to $[0, 1]$. **Third row:** gaussian noise added to the Gestalt code witch is then rounded and clipped to $[0, 1]$. **Fourth row** $0.25$ scaled gaussian noise added to the position (x,y) code which is then clipped to $[-0.9, 0.9]$

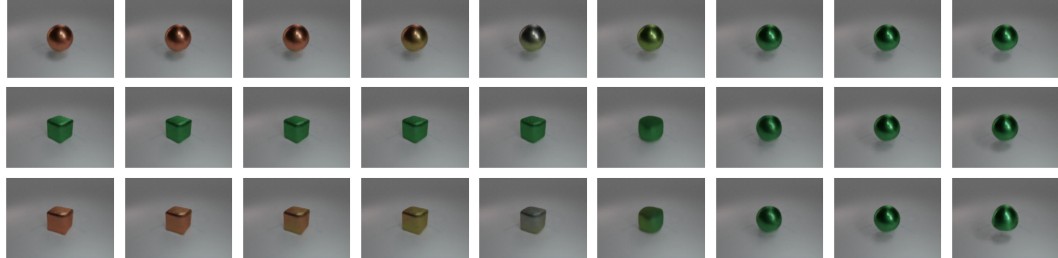

Figure 16: Traversing the Gestalt code manifold: Using a Gaussian Mixture together with t-SNE we find distinct clusters in the latent Gestalt code. Traversing the main axis of variance given by a PCA we found a color axis (**top rob**) and a shape axis (**middle row**). Since these axes are disentangled they can be used in an additive manner to simultaneously change the color and shape (**bottom row**)

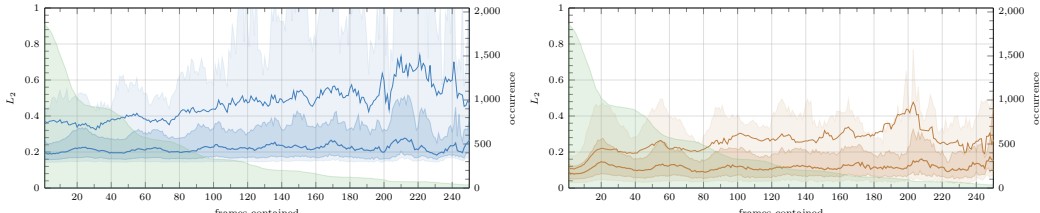

Figure 17: Comparison between unsupervised (**left**) and supervised (**right**) tracking, while the Snitch is contained progressively longer. The x-axis depicts the number of frames the Snitch has been contained. The left y-axis shows the $L_2$ distance to the target position, while the right y-axis shows the number of times a Snitch was contained that long within the test-set (green shaded area). The thick line represents the median $L_2$ distance, while the thin line represents the mean $L_2$ distance. The weaker colored area shows the 90-10-quantile, while the stronger one corresponds to the 75-25-quantile.

## A.4 GESTALT CODE INVESTIGATION

Here we investigate the latent landscape of the Gestalt codes learned by Loci when trained on the CATER challenge. To do this, we first selected a fixed amount (450) Gestalt codes created by Loci using the CATER dataset. We then clustered these Gestalt codes using a Gaussian Mixture model and then performed a dimensionality reduction using t-SNE. In some of the found clusters mainly the color of the objects was different, while in others the color and the shape varied.

The images from Figure 16 where created by using a a Principal component analysis (PCA) to calculate the main axes of variance in different clusters. A random Gestalt code was chosen from a clusters and was modified by subtracting or adding a portion of the axes or in other words, the Gestalt code was used as a starting point and then new Gestalt codes were sampled walking down and up the selected axes. These new Gestalt codes are not binary anymore but were still clipped at 0 and 1, to create meaningful inputs to the decoder. We were able to identify an axis, that only varies the color of the objects, while the shape stays the same, and one axis, which only varies the shape, while the color stays the same. Since these axes are disentangled they can be used in an additive manner to simultaneously change the color and shape of objects.

## A.5 OBJECT PERMANENCE EVALUATION

In order to analysed Loci's object permanence abilities we evaluated how the number of time steps the Snitch was hidden affects Loci's ability to correctly locate it. As shown in Figure 17, where the tracking accuracy is plotted for progressively extended periods of Snitch containment. For both the supervised and unsupervised tracking, the location error remains constant after an initial increase, even over extended time spans.

## A.6 E-PROP FOR GATEL0RD

GateL0RD is defined in Equation 10 till Equation 15 following Gumbsch et al. [39] with the gating network $\mathfrak{g}$, the candidate network $r$, and the cell state $c$. Next, we detail how we applied e-prop in GateL0RD.

### A.6.1 GATEL0RD MODEL

$$\mathfrak{g}_j^t = \Lambda\left(\sum_i \theta_{j,i}^{rec,\mathfrak{g}} c_i^{t-1} + \sum_i \theta_{j,i}^{in,\mathfrak{g}} x_i^t + b_j^{\mathfrak{g}} + \mathcal{N}(0,\Sigma)\right) \qquad \text{(gating network state)} \quad (10)$$

$$r_j^t = \phi\left(\sum_i \theta_{j,i}^{rec,r} c_i^{t-1} + \sum_i \theta_{j,i}^{in,r} x_i^t + b_j^r\right) \qquad \text{(candidate network state)} \quad (11)$$

$$c_j^t = \mathfrak{g}_j^t r_j^t + (1 - \mathfrak{g}_j^t) c_j^{t-1} + h_0^t \qquad \text{(consequent cell state)} \quad (12)$$

$$p_j^t = \phi\left(\sum_i \theta_{j,i}^{rec,p} c_i^t + \sum_i \theta_{j,i}^{in,p} x_i^t + b_j^p\right) \qquad \text{(output factor } p) \quad (13)$$

$$\mathfrak{o}_j^t = \sigma\left(\sum_i \theta_{j,i}^{rec,\mathfrak{o}} c_i^t + \sum_i \theta_{j,i}^{in,\mathfrak{o}} x_i^t + b_j^{\mathfrak{o}}\right) \qquad \text{(output factor } o) \quad (14)$$

$$h_j^t = \mathfrak{o}_j^t p_j^t \qquad \text{(resulting prediction at time } t) \quad (15)$$

### A.6.2 E-PROP UPDATES:

**core derivative and eligiblity determination:**

$$\Lambda' = \frac{\partial \Lambda(x)}{\partial x} = \begin{cases} 0 & \text{if } x \le 0 \\ (1 - \Lambda(x)^2) & \text{otherwise} \end{cases} \qquad \text{(pseudo-derivative of gate)} \quad (16)$$

$$\epsilon_{j,i}^t = \frac{\partial c_j^t}{\partial c_j^{t-1}} \epsilon_{j,i}^{t-1} + \frac{\partial c_j^t}{\partial \theta_{j,i}} = (1 - \mathfrak{g}_j^t)\epsilon_{j,i}^{t-1} + \frac{\partial c_j^t}{\partial \theta_{j,i}} \qquad \text{(cell-specific eligibility value)} \quad (17)$$

**gating network forward eligibility propagation:**

$$\frac{\partial E}{\partial \theta_{j,i}} = \frac{\partial E}{\partial c_j^t}\epsilon_{j,i}^t + \lambda \mathcal{H}(\mathfrak{g}_j^t)\frac{\partial \mathfrak{g}_j^t}{\partial \theta_{j,i}} \qquad \text{(actual update signal)} \quad (18)$$

$$\epsilon_{j,i}^{rec,\mathfrak{g},t} = (1 - \mathfrak{g}_j^t)\epsilon_{j,i}^{t-1} + \Lambda' c_i^{t-1}(r_j^t - c_j^{t-1}) \qquad \text{(recurrent weights-specific elig.)} \quad (19)$$

$$\epsilon_{j,i}^{in,\mathfrak{g},t} = (1 - \mathfrak{g}_j^t)\epsilon_{j,i}^{t-1} + \Lambda' x_i^t(r_j^t - c_j^{t-1}) \qquad \text{(input weights-specific elig.)} \quad (20)$$

$$\epsilon_{j,i}^{b,\mathfrak{g},t} = (1 - \mathfrak{g}_j^t)\epsilon_{j,i}^{t-1} + \Lambda'(r_j^t - c_j^{t-1}) \qquad \text{(bias weight-specific elig.)} \quad (21)$$

**candidate network forward eligibility propagation:**

$$\frac{\partial E}{\partial \theta_{j,i}} = \frac{\partial E}{\partial r_j^t}\epsilon_{j,i}^t \qquad \text{(actual update signal)} \quad (22)$$

$$\epsilon_{j,i}^{rec,r,t} = (1 - \mathfrak{g}_j^t)\epsilon_{j,i}^{t-1} + (1 - (r_j^t)^2)c_i^{t-1}\mathfrak{g}_j^t \qquad \text{(recurrent weights-specific elig.)} \quad (23)$$

$$\epsilon_{j,i}^{in,r,t} = (1 - \mathfrak{g}_j^t)\epsilon_{j,i}^{t-1} + (1 - (r_j^t)^2)x_i^t\mathfrak{g}_j^t \qquad \text{(input weights-specific elig.)} \quad (24)$$

$$\epsilon_{j,i}^{b,r,t} = (1 - \mathfrak{g}_j^t)\epsilon_{j,i}^{t-1} + (1 - (r_j^t)^2)\mathfrak{g}_j^t \qquad \text{(bias weight-specific elig.)} \quad (25)$$

Note that the partial derivatives $\frac{\partial E}{\partial \theta_{j,i}}$ in Equation 18 and Equation 22 address the current time step $t$ only. Also note that $\frac{\partial E}{\partial c^t}$ and $\frac{\partial E}{\partial r^t}$ carry true gradient information of the current time step, backprogated through the feedforward connections of the architecture. In contrast, original e-prop uses local approximations of the learning signal only, which does not allow to stack multiple layers without losing the exact local gradient.

Table 5: Evaluations of Loci's foreground segmentation masks. Trained networks from the Cater challenge are evaluated on CLEVR with mask by running the single CLEVR images 30 iterations through the network and then comparing the masks using the Intersecion over Union (IoU) metric. We compare a mean per mask accuracy (mask avg) and size weighted average that represents how many pixels where segmented correctly (pixel avg).

| Network | mask avg (%) | pixel avg (%) |
|---|---|---|
| 1 | 83.3 | 96.8 |
| 2 | 84.6 | 97.0 |
| 3 | 84.3 | 97.1 |
| 4 | 84.1 | 96.7 |
| 5 | 85.6 | 97.2 |

## B  LOCI ALGORITHM

Loci processes a sequence of RGB video-encoding images $I \in \mathbb{R}^{T \times H \times W \times 3}$. Processing is mostly done slot-wise, whereby the system is initialized with a variable number of $K$ processing slots. Its main components consist of a *slot-wise encoder*, a *transition module*, and a *slot-wise decoder*. Moreover, a background processing module is implemented. The slot-wise encoder is implemented by a tree-structured ResNet-based processing encoder (see Figure 18). The transition module processes *slot-wise temporal dynamics* and *between-slot interaction dynamics* (see Figure 19). The slot-wise decoder is again implemented by a ResNet (see Figure 20). For simple backgrounds, we use a Gaussian Mixture Model to obtain a default background estimate $\hat{R}_{bg}$, which is used for the whole training set. For more complex backgrounds we use an additional Auto-Encoder Module.

In the remainder, we denote scalar values by lower-case letters, tensors by upper-case letters, and vectors by bold letters. Moreover, we denote slot-specific activities with a subscript $k \in 1, .., K$ and time by the superscript $t$. We drop $t$ for temporary values.

We now first define data and neural encoding sizes and types used throughout Loci's processing pipeline. We then specify neural activity initialization. Finally, we detail the unfolding overall processing loop.

### B.1  SLOT-WISE ENCODER

**Inputs**    The encoder inputs at each time step $t$ consist of:

- RGB input image $I^t \in \mathbb{R}^{H \times W \times 3}$,
- MSE map $E^t \in \mathbb{R}^{H \times W \times 1}$ (pixel-wise mean squared error between $I^t$ and $\hat{R}^t$),
- Slot-specific RGB image reconstructions $\hat{R}_k^t \in \mathbb{R}^{H \times W \times 3}$,
- Slot-specific mask predictions $\hat{M}_k^t \in \mathbb{R}^{H \times W \times 1}$,
- Slot-specific mask complements $\hat{M}_k^{t,s} = \sum_{k' \in \{1,..,K\} \setminus k} \hat{M}_{k'}^t$
- Slot-specific isotropic Gaussian position map predictions $\hat{Q}_k^t \in \mathbb{R}^{H \times W}$,
- Slot-specific Gaussian position map complements $\hat{Q}_k^{t,s} = \sum_{k' \in \{1,..,K\} \setminus k} \hat{Q}_{k'}^t$
- Background mask $\hat{M}_{bg}^t \in \mathbb{R}^{H \times W}$, which is equivalent to $1 - \sum_{k \in \{1,..,K\}} \hat{M}_k^t$

**Outputs**    Based on these inputs, the slot-wise encoder network generates latent codes, which are forwarded to the transition module:

- Slot-specific Gestalt codes $G_k^t \in \mathbb{R}^{D_g}$,
- Slot-specific position codes $P_k^t \in \mathbb{R}^4$ encode an isotropic Gaussian $(\boldsymbol{\mu}_k^t, \sigma_k^t)$ and a slot-priority code $z_k^t$,

where $D_g$ denotes the size of the Gestalt code and $\boldsymbol{\mu}_k^t \in \mathbb{R}^2$.

## B.2 TRANSITION MODULE

A transition module is used to process interaction dynamics within and between these slot-respective codes and creates a prediction for the next state, which is fed into the decoder. The input to the transition module equals $G^t$, $P^t$. It is processed across slots and per slot in the respective layers: Multi-Head Attention predicts slot interactions (across slots), while GateL0RD predicts slot-specific dynamics (per slot). In our main CATER implementation we use two attention layers with two heads each with GateL0RD layers in between.

**Outputs**   The outputs of the transition module $\hat{G}_k^{t+1}$ and $\hat{P}_k^{t+1}$ have the same size as its inputs. Additionally, recurrent, slot-respective hidden states $\hat{H}_k^t$ are maintained in the time-recurrent GateL0RD layers:

- Slot-specific position codes $\hat{P}_k^t \in \mathbb{R}^4$,

- Slot-specific Gestalt codes $\hat{G}_k^t \in \mathbb{R}^{D_g}$,

- Slot-specific GateL0RD-layer-respective hidden states $\hat{H}_k^t \in \mathbb{R}^{D_h}$,

where $D_h$ denotes the size of the recurrent latent states.

## B.3 SLOT-WISE DECODER

**Inputs**   The outputs of all slots from the transition module $\hat{P}^{t+1}$ and $\hat{G}^{t+1}$ then act as the input to the decoder.

**Outputs**   The output of the decoder includes the slot-respective masks and RGB reconstructions:

- Slot-specific mask outputs $\hat{M}_k^{t+1} \in \mathbb{R}^{H \times W}$,

- Slot-specific RGB image reconstructions $\hat{R}_k^{t+1} \in \mathbb{R}^{H \times W \times 3}$,

which are then used as part of the input at the next iteration as specified above.

We generate the combined reconstructed image $\hat{R}^{t+1}$ by summing all slot estimates $\hat{R}_k^{t+1}$ and the background estimate $\hat{R}_{bg}$ weighted with their corresponding masks $\hat{M}_k^{t+1}$ and $\hat{M}_{bg}^{t+1}$, as specified further in Algorithm 1.

## B.4 SEQUENCE INITIALIZATION

At time step $t = 1$ we determine the network's inputs based on randomly generated, fictive position and Gestalt estimates for each slot $k$, which are then fed through the Loci decoder module. Initial position and Gestalt codes $\hat{P}_k^1$ and $\hat{G}_k^1$ are sampled from an isotropic Gaussian distribution $\mathcal{N}_{Position}(\boldsymbol{\mu}_p, \sigma_p)$ and a factorized Gaussian distribution $\mathcal{N}_{Gestalt}(\boldsymbol{\mu}_g, \boldsymbol{\sigma}_g)$ with learnable parameters $\boldsymbol{\mu}_p \in \mathbb{R}^3$, $\sigma_p \in \mathbb{R}$ and $\boldsymbol{\mu}_g \in \mathbb{R}^{D_g}$, $\boldsymbol{\sigma}_g \in \mathbb{R}^{D_g}$, respectively. The third position code value of $\hat{P}_k^1$, that is, the Gaussian standard deviation $\hat{\sigma}_k^1$, is set to $1/\texttt{width}$, where $\texttt{width}$ denotes the number of pixels in a row, effectively setting $\hat{\sigma}_k^1$ to one pixel distance. The fourth position code value, that is, the priority value $\hat{z}_k^1$, is set to its index value $\hat{z}_k^1 \leftarrow k$, inducing an ordered priority, which biases initial random slot assignments and thus bootstraps initial slot-assignment progress.

Based on the initial codes, we generate estimates of the output mask $\hat{M}_k^1$, reconstruction $\hat{R}_k^1$ and Gaussian positions $Q_k^1$ by calling the slot-wise decoder (see Algorithm 1 line 19 and following): $\hat{M}_k^1, \hat{R}_k^1, Q_k^1 \leftarrow SlotWiseDecoder(\hat{P}_k^1, \hat{G}_k^1)$. We finally determine the first RGB image reconstruction $\hat{R}^1$.

The hidden states of the recurrent neural network GateL0RD are initialized to zero, that is, $H_k^1 \leftarrow 0$.

## B.5 MAIN PROCESSING LOOP

Loci first runs the main processing loop for ten time steps with the first video image of a particular image sequence. It thereby bootstraps the objects into individual slots, somewhat similar to previous slot-attention approaches [60]. After the ten initial time steps, Loci keeps re-initializing positions $P_k^t$ of a slot $k$ to random values (as specified above) given that $max(\hat{M}_k^t)$ has been smaller than $0.5$ for all time points until $t$. This induces an initial active search process. For longer video sequences, such as the Aquarium footage, this search process was also used for invisible slots, which increased the number of tracked objects, but negatively influenced object permanence.

Note that the prediction error is calculated across the three RGB channels. It determines the MSE between the input $I^t$ and the static background $R_{bg}$, multiplied per Hadamard-product with the forth square root MSE between the input $I^t$ and the predicted input $\hat{R}^t$. The fourth square root emphasizes small differences, encouraging accurate encodings of individual entities.

In the transition module we apply a single trainable parametric bias neuron alpha, as proposed in Bachlechner et al. [1], instead of layer-normalization. Alpha is initialized to zero. Its current

---

**Algorithm 1** `Loci-Algorithm` (main processing loop)

---

1: **Inputs:** Input video $I \in \mathbb{R}^{T \times H \times W \times 3}$, static background $\hat{R}_{bg} \in \mathbb{R}^{H \times W \times 3}$
2: **Network parameters:** $\Theta_{encoder}, \Theta_{transition}, \Theta_{decoder}$
3: **Additional parameters:** initialization parameters $\Theta_{init}$; background threshold $\theta_{bg}$, which is encoded as a uniform offset mask $\tilde{M}_{bg} \leftarrow \theta_{bg}$; number of slots $K$; processing steps $T$

---

4: Initialize $H_k^1, \hat{R}_k^1, \hat{R}^1, \hat{M}_k^1, \hat{Q}_k^1$ # see Section B.4 for details
5: **for** $t = 1 \dots T$ **do**
6:      # Pre-processing:
7:      $E^t \leftarrow \sqrt{\texttt{Mean}\left(\left(I^t - R_{bg}\right)^2, axis = \text{'rgb'}\right)} \circ \sqrt[4]{\texttt{Mean}\left(\left(I^t - \hat{R}^t\right)^2, axis = \text{'rgb'}\right)}$
8:      `compute complements` $\hat{M}_k^{t,s}, \hat{Q}_k^{t,s}$
9:      # Slot-wise encoder:
10:      $S_k^t \leftarrow Encoder_{Trunk}(I^t, E^t, \hat{R}_k^t, \hat{M}_k^t, \hat{M}_k^{t,s}, \hat{Q}_k^t, \hat{Q}_k^{t,s}, \hat{M}_{bg}^t)$
11:      $G_k^t \leftarrow Encoder_{Gestalt}(S_k^t)$
12:      $P_k'^t \leftarrow Encoder_{Position}(S_k^t)$
13:      $P_k^t \leftarrow \texttt{concat}\left[\boldsymbol{\mu}_k^t, \sigma_k^t, z_k^t\right] \leftarrow \texttt{concat}\left[Encoder_{\boldsymbol{\mu}}(P_k'^t), Encoder_{\boldsymbol{\sigma}}(P_k'^t), Encoder_z(P_k'^t)\right]$

14:      # Transition module:
15:      $\hat{G}^{t+1}, \hat{P}^{t+1}, H^{t+1} = TransitionModule(G^t, P^t, H^t)$
16:      # Gestalt binarization:
17:      $\hat{G}^{t+1} \leftarrow \texttt{Sigmoid}(\hat{G}^{t+1})$
18:      $\hat{G}^{t+1} \leftarrow \hat{G}^{t+1} + \hat{G}^{t+1}(1 - \hat{G}^{t+1})\mathcal{N}(0, 1)$
19:      # Slot-wise decoder:
20:      # $p$ encodes all pixel positions normalized to $[-1, 1]$, `width` the number of pixels in a row
21:      $\hat{Q}_k^{t+1} \leftarrow \exp\left(\frac{-(p - \hat{\boldsymbol{\mu}}_k^{t+1})^2}{2\max\left(\frac{1}{\texttt{width}}, \hat{\sigma}_k^{t+1}\right)^2}\right)$
22:      $decode_k \leftarrow \texttt{Priority-Based-Attention}(\hat{G}_k^{t+1}, \hat{Q}_k^{t+1}, \hat{\mathbf{z}}^{t+1})$
23:      $\hat{R}_k^{t+1}, \tilde{M}_k^{t+1} \leftarrow Decoder_{Trunk}(decode_k)$
24:      # Post-processing:
25:      $[\hat{M}_1^{t+1}, \dots, \hat{M}_K^{t+1}, \hat{M}_{bg}^{t+1}] \leftarrow \texttt{softmax}(\texttt{concat}[\tilde{M}_1^{t+1}, \dots, \tilde{M}_K^{t+1}, \tilde{M}_{bg}], axis = \text{'K'})$
26:      $\hat{R}^{t+1} \leftarrow \texttt{sum}(\texttt{concat}\left[\hat{R}_1^{t+1}, .., \hat{R}_K^{t+1}, \hat{R}_{bg}\right] \circ \hat{M}^{t+1}, axis = \text{'K'})$
27: **end for**
28: **return** $[\hat{R}^1 \dots \hat{R}^T]$

---

---

**Algorithm 2** `Priority-based-Attention`

---

1: **Inputs:** Gestalt: $G_k \in \mathbb{R}^{1,D_g}$, Gaussian 2d position: $Q_k \in \mathbb{R}^{H' \times W' \times 1}$, priority: $\boldsymbol{z} \in \mathbb{R}^K$
2: **Additional parameters:** values of the learnable $\boldsymbol{\theta^w} \in \mathbb{R}^K$ are initially set to 25, while $\boldsymbol{\theta^b} \in \mathbb{R}^K = \{0, 1, \dots, (K-1)\}$ induces a default slot-order bias.

---

3: $\boldsymbol{z'} \leftarrow (\boldsymbol{z} \cdot K + \mathcal{N}(0, 0.1) + \boldsymbol{\theta^b}) \circ \boldsymbol{\theta^w}$ # Scale priorities and add Noise

4: # Subtract Gaussian attention from other slots, scaled by priority ($\sigma$ denotes the sigmoid)
5: $Q'_k \leftarrow \max(0, Q_k - \sum_{k' \in \{1,\dots,K\} \setminus k} \sigma(z'_k - z'_{k'}) \cdot Q_i)$
6: $combine_k \leftarrow Q'_k \times G_k$ # $combine_k \in \mathbb{R}^{H' \times W' \times D_g}$

7: **return** $combine_k$

---

value is multiplied with the output vector before the residual parts of the transition module. These alpha-residuals enforce the predictor to initially compute the identity function. The mechanism bootstraps Loci's learning progress by initially focusing it on developing decoder-suitable Gestalt encodings.

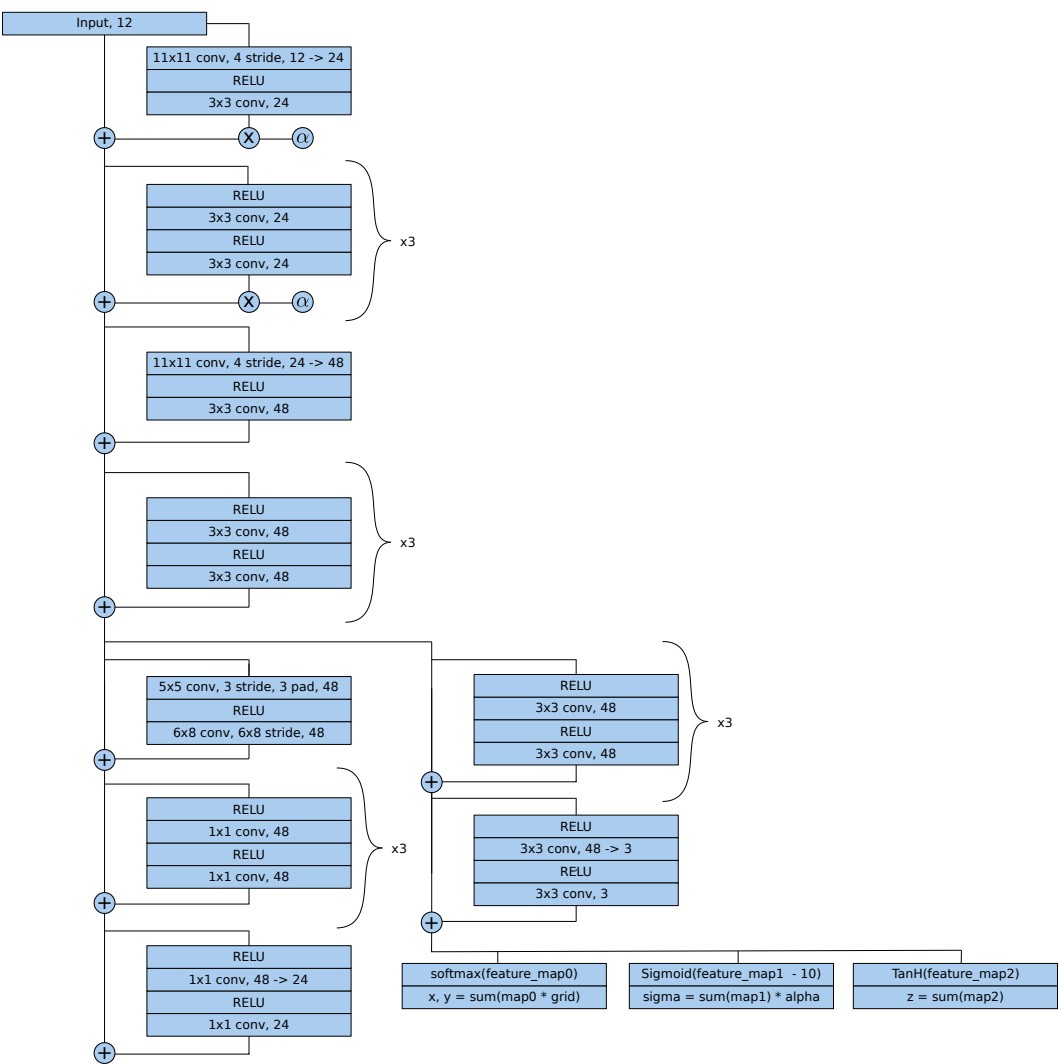

Figure 18: Encoder module diagram (identical for all slots $k$)

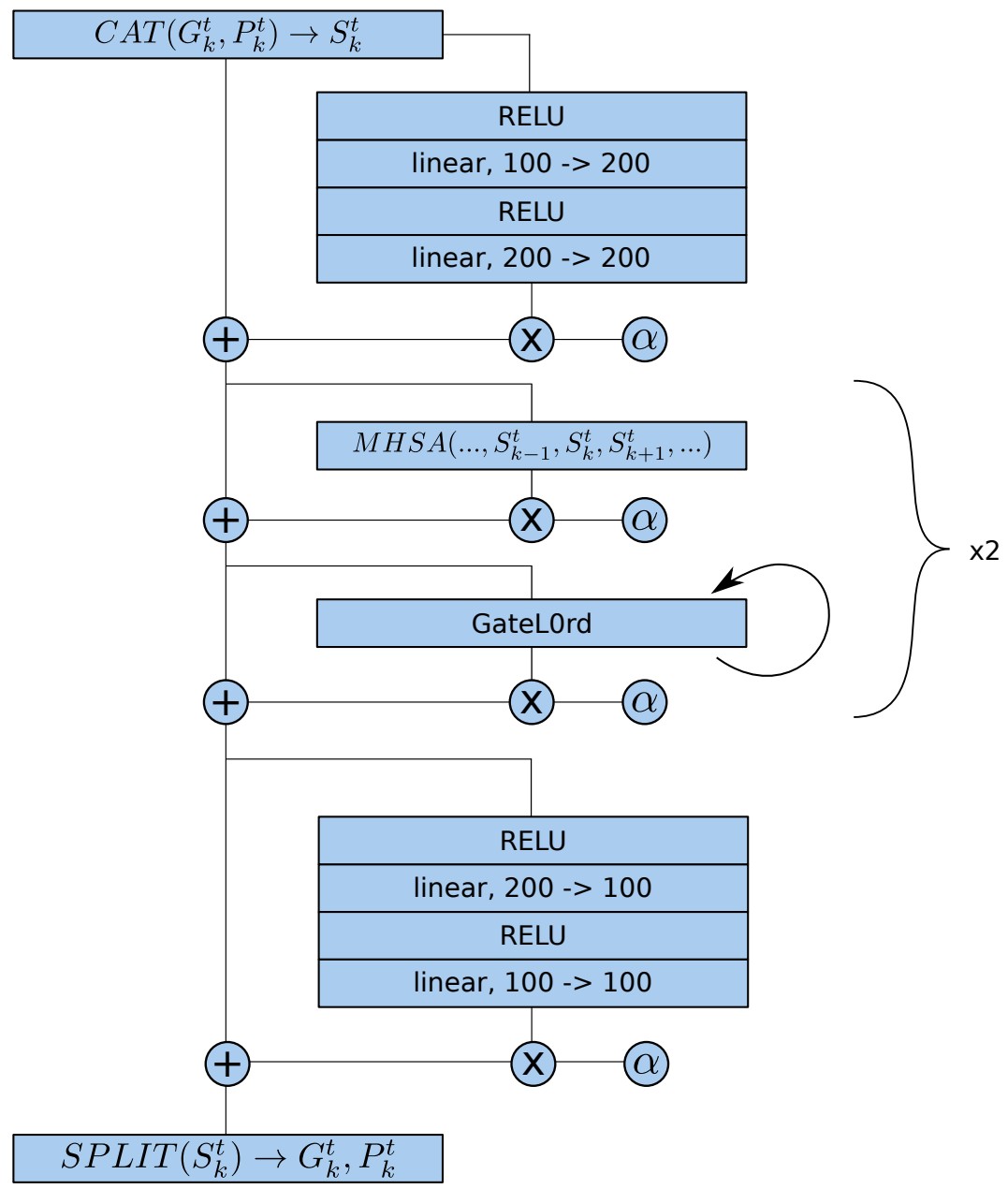

Figure 19: Transition module diagram (identical for all slots $k$)

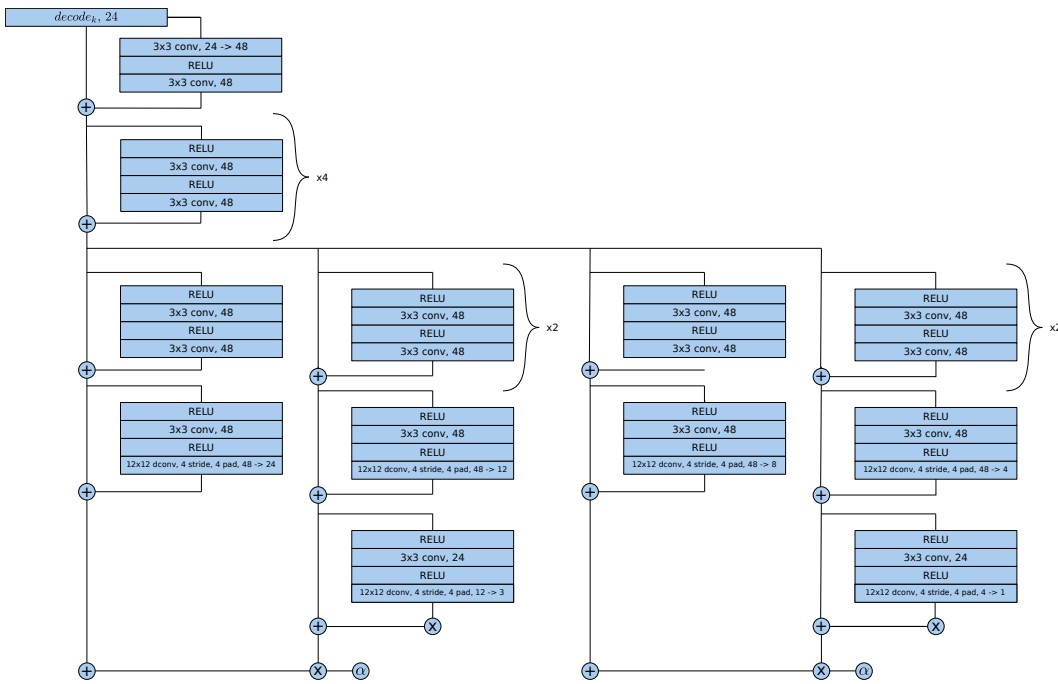

Figure 20: Decoder module diagram (identical for all slots $k$)

## C    POTENTIAL NEGATIVE SOCIETAL IMPACTS

Here we take potential negative social impacts of Loci into consideration. Orientated on the NeurIPS Ethics Guidelines we identified two topics where Loci could potentially be used in an harmful way:

***Could Loci directly facilitate injury to living beings?*** At its current stage it is unlikely that Loci could directly be used in any weapons systems. However, when developing Loci further, its unsupervised nature allows training on huge amounts of unlabeled data and then fine tune a weapons systems to identify and track a specific target.

***Could Loci help develop or extend harmful forms of surveillance?*** Potentially yes; Loci already works well with static cameras and backgrounds. While we did not evaluate Loci's performance on pedestrians, it is likely that Loci could be trained to track pedestrians in a surveillance setting. Supervised fine-tuning could then potentially be used to identify a specific target group of people.

While these potential misuses of Loci are concerning, they are an unavoidable byproduct of advancing the field of (unsupervised) object tracking as a whole. Seeing that current object tracking systems are mostly black boxes, particularly when it comes to deciphering how and why they track certain entities, Loci may enable better control over what is tracked and where exactly tracking may be applied. Hopefully, Loci can thus be used to mitigate unwanted tracking biases, or, at least, to facilitate the identification of such tracking biases.

