# OpenReview forum: "Learning What and Where: Disentangling Location and Identity Tracking Without Supervision"
_ICLR.cc/2023/Conference — ICLR 2023 poster_

### Official Review · Reviewer_fFzL · 2022-10-21

**Confidence:** 4
**Correctness:** 3
**Technical Novelty And Significance:** 2
**Empirical Novelty And Significance:** Not applicable
**Recommendation:** 6

**Clarity, Quality, Novelty And Reproducibility:**

- I had a hard time understanding how the Loci system works and what the specific novel ideas were behind it. Section 3 was hard to parse in particular. I hoped Figures 1 and 2 would illuminate this, but they did not.

- As a result, my understanding is that the Loci system as a whole only makes for an incremental contribution (e.g., as mentioned under Weaknesses), although it is interesting as a "demo" of some promising research topics (slot-based object-centric representations, object-centric dynamics, disentanglement of position and identity).

- Due to the complexity of the architecture and various losses, re-implementing this system from scratch would be difficult.

Misc suggestions
=======
- Title: typo? “Learning What and Where - Unsupervised Disentangling *of* Location and Identity Tracking”?

-  I would suggest spending time improving the last paragraph of the introduction (or adding another paragraph after it). For example, instead of starting this paragraph with “Our goal is to…” and telling the reader what you’d *like to do*, it is usually best to directly tell the reader what you *did* (e.g., a paragraph beginning with “In this work we…”).

- I found it overwhelming to try to parse all the aspects of the Loci system as presented (particularly Section 3). More details are provided in appendix Section B (which I skimmed through). It would help to identify one or two key architectural components that are novel and that “make all the difference”, and to focus the paper on these alone (while simplifying the rest of the system). Maybe more focus should be placed on Loci's slot design --- "each slot has a spotlight"/"tree-structured slots" (??) Overall, this will help a lot with clarity, and then the ablation studies for the one or two new components can be included in the main experimental results to further validate them.

- Having one high-level figure showing how Loci processes an entire sequence of images could be helpful to understand what the inputs and outputs are, how the transition module is used, etc.


**Strength And Weaknesses:**

Strengths
=======
- Identifying how to use self-supervised learning to solve binding-problem-respective challenges for object-centric perception is an important and relevant problem.

- Good empirical results on CATER and MovingMNIST. The results of the self-supervised version are  mostly comparable to baselines while the supervised version shows a stronger improvement.

Weaknesses
=======
- I believe that the idea of disentangling “what” from “where in object-centric representation learning has been explored repeatedly in the past by previous systems (as the authors also indicate in the paper), and so I view this aspect of the paper to be of limited novelty.

- It is unclear to me what other aspects of the Loci architecture are novel. I struggle to see how Loci advances our understanding of how to solve binding-problem-respective challenges. Rather, Loci as a whole appears to be an agglomeration of various ideas from prior work into a single, complex system.


**Summary Of The Paper:**

This paper introduces Loci, a self-supervised system for object-centric location and identity tracking in video. The motivation is to build a hybrid symbolic + neural system that addresses challenges pertaining to the binding problem. Emphasis is placed on disentangling “what” from “where” in each object representation. The main experimental results compare Loci against competitive baselines on CATER and Moving MNIST.


**Summary Of The Review:**

I believe the paper, in its current state, is probably not ready for publication. I encourage the authors to spend time improving the presentation and clarity of the work to be able to identify and highlight novel insights that advance our understanding of how to address aspects of the binding problem. This may involve simplifying the proposed Loci system.

---
Update: After reading the author response, I have increased my score from 5 to a 6. Justification is provided in the comments.

---

> ### Author Response · Authors · 2022-11-16
> **Answer to Reviewer fFzl**
>
> Thank you very much for acknowledging our empirical results and the relevance of the problem Loci is dealing with.
>
> We are sorry that our Figures 1 and 2 did not help you to understand the process pipeline. In order to improve comprehensibility, we have modified Figure 1 substantially, by incorporating your suggestion to add a high-level component, and hope it contributes now better to understanding and parsing Loci's information flow. If you still spot unclear aspects, we are happy to hear any suggestions on how to improve it further.
>
> Please find our response about your concerns relating to the novelty of the separation of ventral and dorsal pathways in the answer to all reviewers above. Since the synergy of all components eventually leads to the successful separation of \`what' and \`where', we decided to maintain the description of the different modules in the main text, hopefully being better supported by Figure 1 now.
>
> In any case, and for the purpose of re-producibility, we will publish the entire model code. We hope that these modifications satisfy the reviewer's concerns related to a potentially challenging re-implementation of our model - and indeed enable such a re-implementation and the direct usage of our code.
>
> We intentionally had formulated the title ambiguously to give way to multiple valid interpretations. However, seeing that it can be misleading, we have modified it to \`Learning What and Where - Disentangling Location and Identity Tracking Without Supervision' in order to reduce ambiguity and to emphasize the primary contribution.

---

> > ### Comment · Reviewer_fFzL · 2022-11-18
> > **Thanks**
> >
> > Thanks for your response and for improving the paper based on reviewer feedback. I believe the improvements to the presentation has increased the readability somewhat. The additions to the introduction also help clarify the significance and novelty of Loci a little.
> >
> > My understanding of Loci remains that it is mainly a well-engineered combination of various architectural designs into a single complex model. While, to me, this means the novelty of the methodological contribution is minor, overall I can see that this is an interesting architecture for studying the disentanglement of what from where. I have increased my score to a 6 to reflect this.

---

### Official Review · Reviewer_Nkjj · 2022-10-23

**Confidence:** 3
**Correctness:** 4
**Technical Novelty And Significance:** 3
**Empirical Novelty And Significance:** 3
**Recommendation:** 6

**Clarity, Quality, Novelty And Reproducibility:**

The paper is well structured and written, except the part which illustrates the neural architecture which is less detailed and difficult to read. This might have an impact o reproduce the results. My other main concern is the claim that the idea to disentangle appearance and position is new. This might be true for this specific field of slot-based generative model learning but this core idea has been reported in other literature earlier (Denil et al., Learning where to Attend with Deep Architectures for Image Tracking, 2011; M. Ranzato, On learning where to look, 2014; Kahou et al., RATM, 2016).

**Details Of Ethics Concerns:**

Tracking poses privacy issues. I would not see any immediate harm to people and society due to this work.

**Strength And Weaknesses:**

The paper follows a line of research with the new idea to disentangle appearance encoding from position encoding. The idea is motivated by the human visual system which also operates as dorsal and ventral streams. The work shows appealing results on the standard datasets (Table 1). There are two shortcomings: 1) the description of the algorithms is somehow difficult to read and understand. My suggestions would be instead of showing pseudo code (page 25) and narrow, nearly unreadable network illustrations (page 3, page 26, 27) to save page space by neglecting the pseudo code and referring to the implementation code. By that the architectural illustration could be made readable which I think is essential to understand the paper. 2) You state on page 3: "But optical flow based methods notoriously struggle with occlusions as they can only represent parts of the scene that are currently visible.". Optical flow by definition is the vector field of apparent motion. So there is no direct intention to represent flow vectors caused by hidden objects in the vector field. I agree OF struggles with occlusion but because they usually do not consider the concept of occlusion. There are exceptions (see e.g. Detecting Occlusions as an Inverse Problem, Estellers & Soatto, J Math Imaging Vis, 2016).

**Summary Of The Paper:**

The paper proposes a new method along slot based, generative multi-object tracking following the approach of representation, segregation and composition (Greff et al. 2020). It claims the idea to introduce two encoding streams one for appearance and one for position. The paper is well written, however some illustrations and the way how the algorithms are described make the paper challenging to understand.


**Summary Of The Review:**

There is a huge effort in the paper showing appealing results on the CATER dataset. The reason the authors claim is the new idea of disentangling appearance from position encoding in their slot network. It seems that this idea is new to this line of research, however it is not new in ML. The paper needs improvement in the illustrations and in the presentation of the algorithms.

---

> ### Author Response · Authors · 2022-11-16
> **Answer to Reviewer Nkjj**
>
> Thank you for describing our manuscript as being well written and for suggesting improvements of selected sections.
>
> As mentioned in the answer to all reviews above, we have reconceptualized Figure 1 to improve comprehensibility and updated the algorithm's pseudo code accordingly to match the naming conventions. Since we perceive the pseudo code beneficial for the detailed process understanding, we decided to keep it as part of the appendix. Also, we agree that the network illustrations were too small and rearranged them to increase their size (see Figures 18 through 20 in the Appendix). In order to reproduce our experiments and to facilitate the full reconstruction of our architecture, the source code will be made available alongside the publication of the paper.
>
> Although OF is a simple and powerful tool, we agree that it is, per definition and design, not able to handle occlusions. Since we understand the explicit consideration of occlusion substantial for a coherent scene understanding and representation, we advance over the OF by incorporating occlusions explicitly in Loci's processing pipeline in the temporally stable \`what' and \`where' pathways. We added a short statement about the design-based limitation of OF to our manuscript (Section 2, Tracking models).
>
> Unfortunately, we had not been aware of the referenced literature. Thank you for pointing it out to us! The disentanglement of \`what' from \`where' is indeed similar, even though the work focuses on visual attention via an adaptive bounding box (its glimpse) and single object tracking.
> We included the work you have pointed us to and briefly discuss the relations in Section 2 (paragraph Representation).
>
> In respect to the further novelty of the \`what'- and \`where'-separation, please also refer to our answer to all reviewers above.

---

> > ### Comment · Reviewer_Nkjj · 2022-11-23
> > **Changed recommendation to 6**
> >
> > Thanks for your answers to my comments. The illustrations in the appendix are now  readable. I am not happy with this sentence: "But optical flow based methods are not designed to deal with occlusions as they only represent parts of the scene that are currently visible." as there are methods based on OF which detect occlusions either by using the forward and backward OF or by directly evaluating the residual of the objective function. You should again considering rewriting this sentence. OF methods do not describe occlusion explicitly as your approach.
> > The novelty of the paper has been better worked out and described in the paper. This leads me to the conclusion to see the paper above the level of acceptance.

---

### Official Review · Reviewer_BwtU · 2022-10-24

**Confidence:** 3
**Correctness:** 4
**Technical Novelty And Significance:** 3
**Empirical Novelty And Significance:** 3
**Recommendation:** 8

**Clarity, Quality, Novelty And Reproducibility:**

The paper and the proposed method are clear (except for the parts which I'll explain in my comments below). The proposed method is novel, and can have a large impact on both machine learning and cognitive neuroscience.

**Strength And Weaknesses:**

$\textbf{Strengths}$:

The main idea underlying the proposed method is an interesting, algorithmic implementation of the parallel visual processing in the brain. The results, as presented in the paper, strongly support the efficacy of the proposed method. The paper is very well written, and except for one part of the architecture, the rest of the paper is straightforward and easy to understand. Thorough ablation experiments perfectly demonstrate the contribution of each design choice in the performance of the model. Overall, I believe the manuscript is ready for publication as is, and can make a significant contribution to both machine learning and cognitive neuroscience.

$\textbf{Weaknesses}$:

As mentioned above, I believe this paper doesn’t have a significant weakness that needs to be addressed before acceptance. However, there are two parts of the paper that could benefit from further explanation and exploration: the transition module and the information represented by the Gestalt and the position variables. I’ll elaborate in my questions and comments.


**Summary Of The Paper:**

In this paper, inspired by the “what” and “where” parallel pathways of the visual system, the authors present a novel architecture for object segmentation and tracking with disentangled representations of objects and their position. The proposed slot-based object representation architecture excels on the position tracking benchmark CATER which is a complex task requiring detecting and tracking multiple objects and understanding their interactions in dynamic scenes.


**Summary Of The Review:**

$\textbf{Questions}$:

1- How do we know whether the Gestalt and the position encoding variables in the trained model represent the intended information, e.g. object identity and position, respectively? The Gestalt and position lesion results shown in figures 14 and 15 imply a more significant role for the position than the Gestalt. Especially, in the example in figure 15, the Gestalt lesion doesn’t seem to damage shape or object information much in the reconstructed images. How would the lesions affect the moving MNIST task? It should be simpler to see the effects of lesions in the moving MNIST task, e.g. Gestalt lesions should damage digit shapes (or misrepresent digits) while position lesions should interfere with the digits’ movements.

2- Do different slots learn to represent different objects in a scene? Showing a few examples would help.

3- The architecture of the transition module isn’t clear. Unlike the other parts of the model, the transition module's equations are not included in the appendix either. I found it difficult to follow the inner functioning of the transition module. For example, what does “interaction between slot-encoded entities” mean?

4- Although explained in some details in the main text, the results with more complex backgrounds are only limited to one example (the aquarium) in the appendix. Can the authors give more examples of segmentation/training tasks with complex, naturalistic backgrounds? Although the performance of the model is impressive in the case of the given example, the average performance of the model and its cases of failure cannot be understood from the given example.

---

> ### Author Response · Authors · 2022-11-16
> **Answer to Reviewer BwtU**
>
> We appreciate the reviewer's acknowledgement of our work, emphasizing the quality of the presentation, thorough evaluation, and a significant contribution to machine learning and cognitive neuroscience. Thank you very much indeed!
>
> We have updated Figure 1 of our manuscript to better convey the details of the transition module. Moreover, we now provide a close-up schematic in Figure 19 of the appendix, outlining the underlying building blocks that constitute the transition module. Please zoom into the schemes digitally - we tried our best to plot Figure 19 (as well as particularly figures 18 and 20) more compactly and to increase its size.
>
> We now repeated the lesion studies on the moving MNIST dataset and performed the according analysis; results can now be found in figures 13 and 15. Indeed, we agree that the manipulation of the Gestalt code does not seem to affect the metrics as much as the manipulation of the position code (quantitative results in Figures 12 and 13). However, we attribute this to the difficulty to find a reasonable quantitative description of an object's appearance (Gestalt) - an issue that is widely discussed in the literature (cf. e.g. Andonian et al. 2021 for a recent treatise) and which has been addressed in works trying to develop appearance-specific metrics, such as PSNR or SSIM (Wang et al. 2004). To demonstrate the effect of the Gestalt code, we provide a couple of qualitative results in Figures 14 through 16, which vividly confirm your expectation: A modification of the Gestalt code effectively changes the object appearance (conserving the object position), while a change in the position code alters the object's position (leaving the Gestalt untouched).
>
> The interaction between slot-encoded entities relates to the multi-head self-attention module, which can exchange information across slots in order to facilitate their temporal predictions (e.g., a force exchange upon impact). The GateL0RD component does not have the capacity to exchange information across entities, but it is able to process exchanged information (from the multi-head attention) further.
> We hope that the updated Figure 1 makes this more intuitive now.
>
> As detailed in Section 3.2, an important aspect for successful training is the use of a warm-up phase, where we mask the loss of the network with a foreground mask obtained from a pre-trained background module (or pre-calculated, static background image). We currently rely on a very simplistic background autoencoder to learn a sufficiently accurate background representation during pre-training. Hence, in it's current stage, Loci struggles with complex and dynamic backgrounds, which we aim to improve in future work (see also the brief discussion in Section 5.

---

### Official Review · Reviewer_vHnv · 2022-10-25

**Confidence:** 3
**Correctness:** 3
**Technical Novelty And Significance:** 3
**Empirical Novelty And Significance:** 3
**Recommendation:** 8

**Clarity, Quality, Novelty And Reproducibility:**

Clarity and quality: good. Overall the paper is well-written and the results are nicely presented.

Originality: good. I am not an expert in object tracking, but this appears to be a novel architecture for this type of problem.

**Strength And Weaknesses:**

**Strengths**


1. For the most part the paper is well-written and straightforward to follow.
2. While Loci is fairly complex, I found most of the modeling decisions to be reasonably well-justified.
3. The results, especiall on CATER, are impressive.

**Weakness**


1. I found parts of the model to be under-described, e.g. what exactly is a Gestalt code. This should be presented clearly in the main text since it is central to the model.
2. The total loss function appears to be specifically tuned to CATER. For instance, would the position change loss need to be altered to accomodate different object movement speeds?


**Summary Of The Paper:**

Here, the authors desribe a variant of Siamese networks that performs objecting identification and tracking, Loci. A key feature of Loci is that it treats the “Gestalt” of objects separately from their position. Loci is benchmarked using CATER and Moving MNIST, and it generally shows impressive performance gains relative to other techniques, especially in CATER.

**Summary Of The Review:**

This work presents an advance over the current state of the art in object identification and tracking. Moreover, it is clearly presented and well-argued. Overall I think this would be a good contribution to ICLR.

---

> ### Author Response · Authors · 2022-11-16
> **Answer to Reviewer vHnv**
>
> Thanks you so much for your encouraging review and for pointing out the impressive results on CATER and the novelty of our approach.
>
> We share your concern about an insufficient explanation of the Gestalt code and have addressed this in the improved Figure 1 of Loci's processing architecture, the algorithmic description, Section 3, and the additional Gestalt code analysis in Figure 16: Traversing the Gestalt code manifold (see also our response above).
>
> The position loss with its scaling value of $0.01$ is mainly designed to punish the model for swapping slots, which would cause much larger changes in position. Clearly, if the subsequent video frames are distant in time or the objects move excessively fast, the system will not be able to learn. Since the performance will also depend on the complexity of the video material and the training statistics (e.g., the number of objects that move very fast), we leave it to future analyses and system applications to identify the sweet spot, where optimal performance may be achieved with respect to scene reconstruction, temporal prediction, slot-specific object tracking, etc. However, Figure 11: Importance of regularization losses, provides also an ablaton on the impact of this loss in the CATER task.

---

> > ### Comment · Reviewer_vHnv · 2022-11-19
> > **Thank you for addressing my major comments**
> >
> > I thank the authors for addressing both of my comments, the model description is now fairly clear. I think this work will be a meaningful contribution to the field.

---

### Author Response · Authors · 2022-11-16
**Answer to all Reviewers**

First of all, we want to express our appreciation to the reviewers' and area chair's detailed, excitingly positive, and constructive feedback and the time they have spent to improve our work. We worked hard to increase the conciseness and comprehensibility of our manuscript. In the following, we respond to the general points that were addressed by multiple reviewers. Please find our answers to specific questions in the responses to the individual reviews.

A common concern raised by all reviewers was the difficulty in understanding the model description and how the various modules are linked together. As a result, we re-conceptualized the model's illustration, incorporating the provided suggestions (now Figure 1).
In accordance with the modified figure, we have also updated the pseudo code of the main algorithm (Algorithm 1) to match the terminology.
We hope that the new illustration improves the comprehensibility of our model further. Please let us know about additional issues in this respect, we would be happy to improve the model sketch even further.

Furthermore, the Gestalt code and, in particular, what it represents, has been addressed in multiple reviews (vHnv and BwtU). A Gestalt code can intuitively be understood as a symbolic representation (in form of a latent vector) of an object's appearance, including its shape, color, texture, and other kinds of visual properties. It is clearly separated from the spatial properties of the object (position in image, size, and approximate distance), which are modeled in a separate pathway in Loci. We agree that this is a crucial component of our model description and have emphasized this point in Section 3 of our manuscript. Moreover, we have added an analysis of the structure of the Gestalt code, which enabled us to identify main color- and form-manipulating axes, which can flexibly and independently modify a particular object appearance (cf. Figure 16: Traversing the Gestalt code manifold).

Another point that was addressed by reviewers Nkjj and fFzL concerns the novelty of separated pathways. We absolutely agree that the conceptual idea of disentangling \`what' from \`where' is by far not novel and we did not intend to claim this as part of our contributions; cf. our references [61, 72] to the historical works by Mishkin et al. 1983 and Ungerleider et al. 1994 at the end of the introduction. However, what is new are the architectural biases that we implement in order to develop a self-organized, consistent, and successful disentangling. As suggested by reviewer fFzL, we thus also modified our introduction (last part) and attempt to explicate key novelty components. Naturally, ``novelty'' is always slightly ill-defined - thus, we would be happy to receive additional feedback about this passage to potentially identify further associations to the literature, which we may have overlooked.

Overall, we hope the reviewers find the time to examine our manuscript changes. We would appreciate any further feedback and are happy to both incorporate any further suggestions and incorporate further related work / system / process relations. Please find modifications in the .pdf emphasized in color. Note that we did not color minor textual corrections.

---

### Decision · Program_Chairs · 2023-01-20

**Decision:**

Accept: poster

**Justification For Why Not Higher Score:**

The algorithm is only evaluated on synthetic datasets. The approach is a well engineered combination of previous techniques.

**Justification For Why Not Lower Score:**

All reviewers recommended acceptance. While the novelty is a bit limited, and the method is not evaluated on natural datasets, the performance gains are solid on a task that has attracted a lot of work from the community.

**Metareview: Summary, Strengths And Weaknesses:**

 The authors propose a new approach for multiple object identification and tracking. The main novelty in this work is to disentangle the appearance encoding form the position(location) encoding, an idea inspired from the human visual system.  The paper is mostly clearly written, the approach is well motivated, and the empirical results show significant performance gains. All reviewers agree that the paper should be accepted.

**Note From Pc:**

if the above contains the word "oral" or "spotlight" please see: "oral" presentation means -> notable-top-5% and "spotlight" means -> notable-top-25%. As stated in our emails, we are disassociating presentation type from AC recommendations